# Generating combinatorial diversity via engineered V(D)J-like recombination in *Saccharomyces cerevisiae*

Andrew P. Cazier[1], Jaewoo Son[1], Sreenivas Yellayi[1], Lizmarie S. Chavez[1], Caden Young[1], Olivia M. Irvin[1], Hannah Abraham[1], Saachi Dalvi[1] & John Blazeck [1,2,3,4,5] ✉

V(D)J recombination is integral to the development of antibody diversity and proceeds through a complex DNA cleavage and repair process mediated by several proteins, including recombination-activating genes 1 and 2, *RAG1* and *RAG2*. V(D)J recombination occurs in all jawed vertebrates but is absent from evolutionarily distant relatives, including the yeast *Saccharomyces cerevisiae*. As yeast grow quickly and are a platform for antibody display, engineering yeast to undergo V(D)J recombination could expand their applicability for studying antibody development. Therefore, in this work we incorporate *RAG1* and *RAG2* into yeast and characterize the resulting recombination ability using a split antibiotic resistance assay, demonstrating successful homology-assisted formation of coding joints. By pursuing a variety of strategies, we increase the rate of homology-assisted recombination by over 7000-fold, with the best rates approaching 1% recombination after four days. We further show that our platform can assay the severity of several disease-causing *RAG1* mutations. Finally, we use our engineered yeast to simultaneously generate up to three unique fluorescent proteins or two distinct antibody fragments starting from an array of nonfunctional gene fragments, which we believe to be the first-ever generation of genetic and phenotypic diversity solely using random recombination of preexisting DNA in a non-vertebrate cell.

V(D)J recombination is a fundamental process in adaptive immunity, enabling the diversification of both immunoglobulins and T-cell receptors in jawed vertebrates[1]. V(D)J recombination allows the semi-random recombination of gene fragments that encode a subsequence of a T-cell receptor or antibody protein, thereby generating vast combinatorial diversity in the immune receptor repertoire, which is essential to the immune system's ability to recognize diverse antigens[2,3]. V(D)J recombination in B cells allows creation of a diverse antibody repertoire with both junctional and combinatorial diversity[4,5].

It is driven by the recombination-activating genes *RAG1* and *RAG2*, which are thought to have evolved in jawed vertebrates from the domestication of a transposon roughly 500 Mya[6].

The yeast *S. cerevisiae* is frequently used for antibody-fragment isolation and engineering because of their ability to surface display genetically encoded antibody libraries[7,8]. Furthermore, yeast are a well-recognized model eukaryotic organism that has proven useful for studying more complex eukaryotic cellular processes[9,10]. Yeast cannot generate antibody combinatorial diversity in vivo, as they are unable to

[1]School of Chemical and Biomolecular Engineering, Georgia Institute of Technology, Atlanta, GA, USA. [2]Parker H. Petit Institute of Bioengineering and Bioscience, Georgia Institute of Technology, Atlanta, GA, USA. [3]Integrated Cancer Research Center, Georgia Institute of Technology, Atlanta, GA, USA. [4]Georgia Immunoengineering Consortium, Emory University and Georgia Institute of Technology, Atlanta, GA, USA. [5]Winship Cancer Institute, Emory University, Atlanta, GA, USA. ✉e-mail: john.blazeck@chbe.gatech.edu

perform V(D)J recombination. Instead, the combinatorial diversity of yeast-display antibody libraries is often generated inside a mammalian host's B cells (e.g., human, mouse, llama) in vivo, and then this diversity is harvested ex vivo and imported into yeast cells using traditional molecular cloning techniques[11,12]. We hypothesized that a yeast strain that could diversify antibody sequences in a manner similar to B cells could be a helpful tool towards 1) studying the protein components of V(D)J recombination, and 2) generating combinatorial diversity in vivo. Therefore, we sought to engineer yeast with the RAG proteins. Yeast, being evolutionarily distant relatives of jawed vertebrates, lack any homologs to *RAG1* and *RAG2* or their transposon prototypes, and as such, have no ability to perform V(D)J recombination.

To highlight one genetic locus on which V(D)J recombination is performed, in pro-B cells, a full-length heavy chain antibody variable domain is formed through sequential recombination of an array of V, D, and J gene segments, with a D and J subunit being brought together first, followed by fusion of a V gene to the DJ sequence, hence V(D)J recombination[13]. Each recombination event requires precise RAG-mediated DNA cleavage at specific sites adjacent to V, D, or J gene segments, followed by DNA repair, resulting in the excision of intervening genomic DNA regions[1]. The DNA cleavage step requires two RAG1 and two RAG2 proteins to form a heterotetramer complex that binds to recombination signal sequences (RSSs)[14]. RSSs contain largely conserved heptamer and nonamer DNA sequences critical to RAG complex recognition separated by 12-bp or 23-bp spacer regions which are less conserved but still able to influence recombination activity[15,16]. The RAG complex follows the '12/23 rule', meaning it binds one RSS with a 12-bp spacer (12-RSS) and one with a 23-bp spacer (23-RSS) before cleaving DNA[17,18]. The genomic arrays of V, D, and J gene segments are organized and include RSSs such that a combinatorially diverse repertoire of full-length immunoreceptors is formed by immune cells.

The high mobility group box 1 or 2 (HMGB1 or HMGB2) proteins assist with RSS binding and cutting by bending 23-RSS DNA, which can substantially increase the rate of RAG-mediated DNA cleavage and eventual recombination[19,20]. DNA strands are cleaved in two steps. First, the RAG complex nicks DNA at the 5' end of each RSS, adjacent to the heptamer, creating a 3' hydroxyl group. Then, an interstrand transesterification reaction occurs where the hydroxyl group attacks the phosphate on the complementary DNA strand, causing a double strand break (DSB) precisely between the RSS heptamer and the DNA that will encode an immune receptor[21]. This process ultimately results in two DNA ends with hairpins, called the coding ends, and two DNA ends with blunt ends that contain the RSSs, called the signal ends[22].

The hairpinned coding ends must then be opened and repaired via the non-homologous end joining (NHEJ) pathway to allow formation of a coding joint, e.g., the V to DJ genetic fusion that results in a functional immunoreceptor variable region. To form this functional coding joint (and repair the genome), the hairpins are opened by a complex consisting of Ku70, Ku80, DNA-PKcs, and Artemis[23], and then NHEJ-mediated DNA ligation follows, requiring the enzymes XLF, XRCC4, and DNL4[24,25]. In B cells, coding joints often have junctional diversity between the two fused gene segments caused by the hairpin opening and by template-independent addition of nucleotides added by polymerases during NHEJ, such as TdT[26,27]. In addition to coding joint formation, the blunt RSS-containing signal ends of DNA are similarly ligated together to form a signal joint that 1) does not encode an immunoreceptor and 2) has been excised from the genome.

Two prior studies have investigated the possibility of a yeast strain that can perform V(D)J recombination, showing that expression of mouse RAG1 and RAG2 in *S. cerevisiae* allows recombination of DNA bearing RSSs to form signal joints or causes transposition events[28,29]. This important work also confirmed that signal joint formation required yeast NHEJ protein machinery, such as YKU70, LIG4, and the MRX complex (MRE11, RAD50, and XRS2), showing that yeast can

repair signal joints similarly to mammalian cells[28,29]. However, the level of signal joint formation seen in these studies was extremely low (~1 signal joint or 4 transposition events per 10 million cells), requiring detection via a highly sensitive, nested PCR assay of bulk DNA. As such, no cells bearing a recombination event were able to be isolated, and a phenotypic test meant to detect DNA cleavage activity failed because it was below the assay's limit of detection[28]. Lastly, these prior studies did not attempt to detect coding joints.

In this work, we demonstrate RAG-mediated coding joint formation in *S. cerevisiae*. Our yeast-based coding joint formation relies on homology between the coding ends but appears to follow the traditional mechanism of V(D)J recombination. We verify that both RSSs and functional RAG1 and RAG2 proteins are required for this recombination, and we show that recombination activity is controlled in part by their ability to physically localize in the yeast nucleus over the cytoplasm or nucleolus. We increase our initial recombination rate over 7000-fold by applying codon optimization, varying protein combinations, adjusting RAG1 truncation, and optimizing the target substrate—ultimately reaching almost 1% recombination after 4 days. We further explore the lower limits of homology needed to allow our assisted recombination to occur and saw powerful effects when altering the DNA between RSSs. In a separate assay, we form and isolate signal joints, exhibiting another hallmark of V(D)J recombination in yeast. Importantly, we show that our yeast strains can effectively quantify the recombination efficiency of mutant RAG1 genes associated with immunodeficiency. Finally, in two separate demonstrations using split fluorescent proteins and scFvs, we use our system to generate RAG-enabled protein diversity. In this manner, we show in a non-vertebrate host cell the ability to study RAG1-associated immune deficiency and to generate in vivo combinatorial protein diversity via recombination of a predefined genetic locus.

## Results

### Expression and localization of RAG1 and RAG2 in yeast

Based on the extremely low rate of signal joint formation mediated by RAG1 and RAG2 co-expression in yeast previously reported[28], we first analyzed the relative expression and subcellular localization of full-length RAG proteins, as well as their 'core' variants that have been truncated to allow for easier heterologous expression and purification[22,30]. Subcellular localization is a critical regulator of RAG activity in mammalian cells, especially the localization of RAG1 to the nucleolus, which downregulates recombination activity in a pre-B cell model system[31,32]. Mouse RAG proteins and their core versions were C-terminally tagged with eGFP and expressed under the control of strong, galactose-inducible promoters. eGFP-tagged proteins were integrated into yeast strains engineered to express either NAB2-mCherry or NOP56-mCherry fusion proteins, as they allow visualization of the nucleus and nucleolus, respectively[33,34]. We saw that RAG1core had much higher expression than the full-length protein, as gauged by eGFP fluorescence intensity, whereas RAG2 expressed well as either a full-length protein or truncated to its core version (Fig. 1a). The full-length RAG1 protein localized predominantly to the nucleolus, whereas RAG1core localized to the nucleus, consistent with work in mammalian cells (Fig. 1b–e, Supplementary Fig. S1)[31,32]. The full-length RAG2 protein displayed the desired nuclear localization, while RAG2core was dispersed in the yeast cytoplasm (Fig. 1b, c). Neither RAG2 nor RAG2core localized to the nucleolus (Fig. 1e, Supplementary Fig. S2).

We attempted to improve RAG1core and RAG2core nuclear localization by fusing a strong nuclear localization signal (NLS) from histone H3 (HHT1) to the N terminus of each enzyme[35]. Surprisingly, expression was greatly improved for NLS-RAG1core, though the localization shifted away from the nucleus and towards the nucleolus (Fig. 1a–e). In contrast, NLS-RAG2core appeared to decrease in expression relative to RAG2core, but the localization to the nucleus was improved (Fig. 1a–c). Based on the much higher expression and

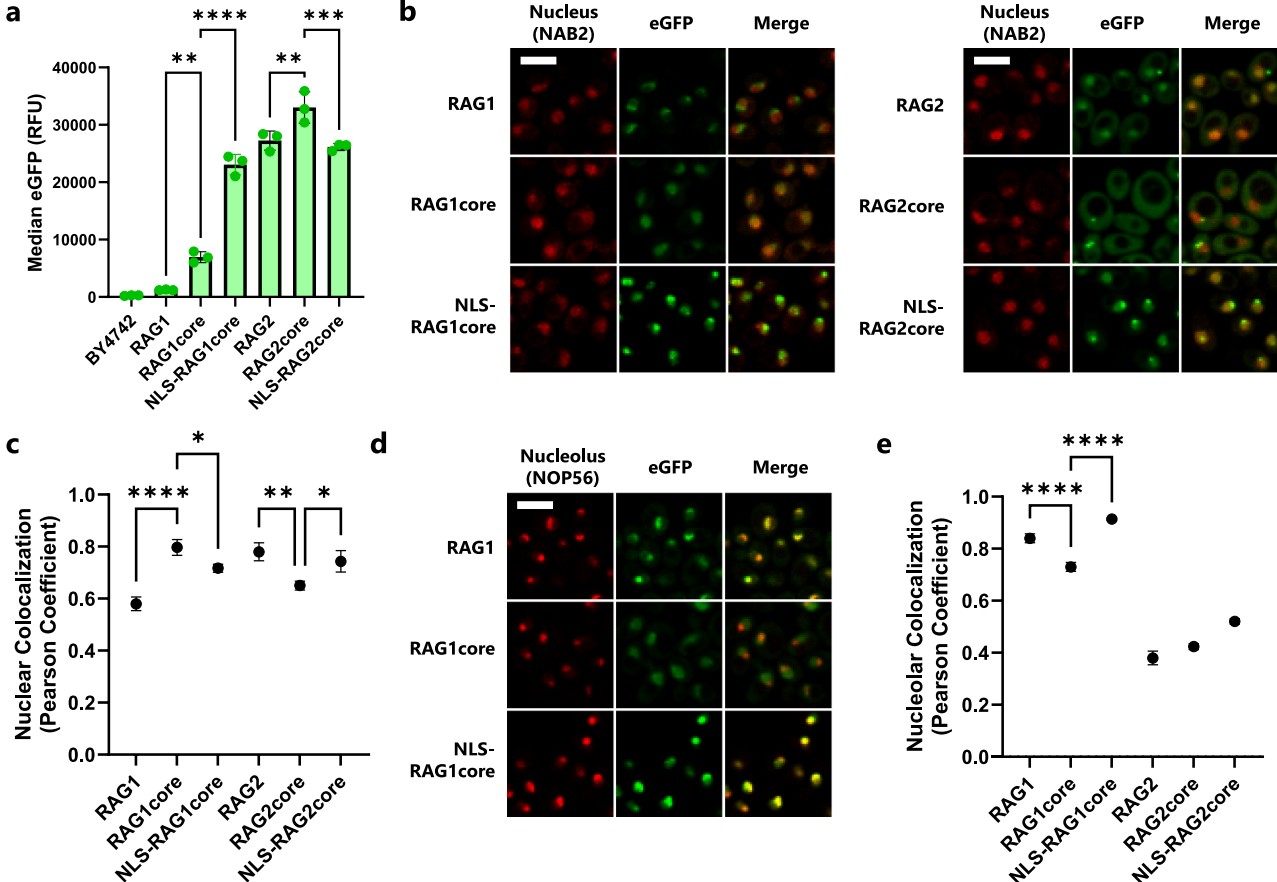

**Fig. 1 | eGFP tagging of RAG1 and RAG2 sheds light on expression and nuclear localization. a** RAG1 or RAG2 variants were tagged with eGFP on the C-terminus and integrated in BY4742-NAB2-mCherry yeast. Median eGFP fluorescence was then measured using flow cytometry after overnight culture in YPG. **b** Confocal microscopy images of eGFP-tagged RAG1 and RAG2 protein variants (green) in a NAB2-mCherry strain which natively highlights the nucleus (red). Scale bar represents 5 μm; all images are at the same scale. Gains have been adjusted from the original images to aid visual comparison. The brightfield channel for each image can be found in Supplementary Fig. S1. **c** Nuclear localization was estimated by calculating the Pearson correlation coefficient between the RAG-eGFP and NAB2-mCherry signals of the confocal images. **d** Confocal microscopy images of eGFP-tagged RAG1

protein variants (green) in a NOP56-mCherry strain which natively highlights the nucleolus (red). As in (**b**), scale bar represents 5 μm. The complete set of images can be found in Supplementary Fig. S2. **e** Nucleolar localization was estimated by calculating the Pearson correlation coefficient between the RAG-eGFP and NOP56-mCherry signals of the confocal images. RFU = relative fluorescence units, NLS = nuclear localization signal. In (**a**, **c**, and **e**) data are presented as mean values ± SD; $n = 3$ biological replicates. Statistical significance was calculated with a one-way ANOVA and Tukey test (*$p < 0.05$, **$p < 0.01$, ***$p < 0.001$, and ****$p < 0.0001$). From left to right, the highlighted $p$ values are, in (**a**), $p = 0.0014$, <0.0001, 0.0012, and 0.0001; in (**c**), $p =$ <0.0001, 0.0499, 0.0016, and 0.0194; and in (**e**), $p =$ <0.0001 and <0.0001. Source data are provided in the Source Data file.

better nuclear localization of RAG1core and NLS-RAG1core relative to full-length RAG1, we expected these variants to outperform RAG1 in recombination. For RAG2, based on the superior localization of full-length RAG2 and NLS-RAG2core compared to RAG2core, we predicted that one of these variants would be optimal for recombination.

## Enabling and improving homology-assisted coding joint formation in yeast

Given that *S. cerevisiae* prefer homology-directed repair (HDR) over NHEJ when resolving DSBs, we sought to make a recombination substrate suited for repair inside a yeast host[36]. We hypothesized that adding a small region of homology (i.e., 20 bp) between two coding ends that would arise after RAG-mediated cleavage would allow the yeast to repair the DSB more effectively without greatly increasing RAG-independent recombination. Therefore, we split a G418 resistance (*G418R*) gene by inserting a 2-kb piece of DNA flanked by a canonical 12-RSS and 23-RSS. The 20 bp at the 3′ end of the first *G418R* fragment was duplicated at the 5′ end of the second fragment (Fig. 2a). The RSSs were oriented to have the heptamers next to the split *G418R* fragments, such that homology-assisted coding joint formation would reform an intact *G418R*. The DNA sequence between the RSSs

contained a URA3 expression cassette and a strong promoter pointed at the 23-RSS—following previous work which has suggested transcriptional activity increases RAG accessibility[28,37]. The final recombination-substrate plasmid was dubbed pY112-CJA-UP-H20.

To begin to gauge the ability of RAG1/2 co-expression to mediate recombination in *S. cerevisiae*, we engineered yeast strains that expressed combinations of the mouse RAG1, RAG2, and (optionally) human HMGB1 proteins. Full-length or core RAG1 and RAG2 variants were placed under control of galactose inducible promoters, while HMGB1 was expressed constitutively if included. Human and mouse HMGB1 sequences are nearly identical, with only two conservative amino acid substitutions differentiating the two (Supplementary Fig. S3). For our initial tests, all proteins were integrated into BY4742 strains that harbored the pY112-CJA-UP-H20 plasmid. Single colonies of each strain were picked, cultured, and spread on G418 antibiotic plates to detect RAG-assisted recombination.

We saw that mouse RAG1core and RAG2core mediated roughly 2- to 4-fold more recombination than the full-length RAG1 and RAG2 proteins, with or without co-expression of HMGB1 (Fig. 2b), and Sanger sequencing of nine colonies showed a seamless recombination of the *G418R* gene (Supplementary Fig. S4a). Low recombination levels seen

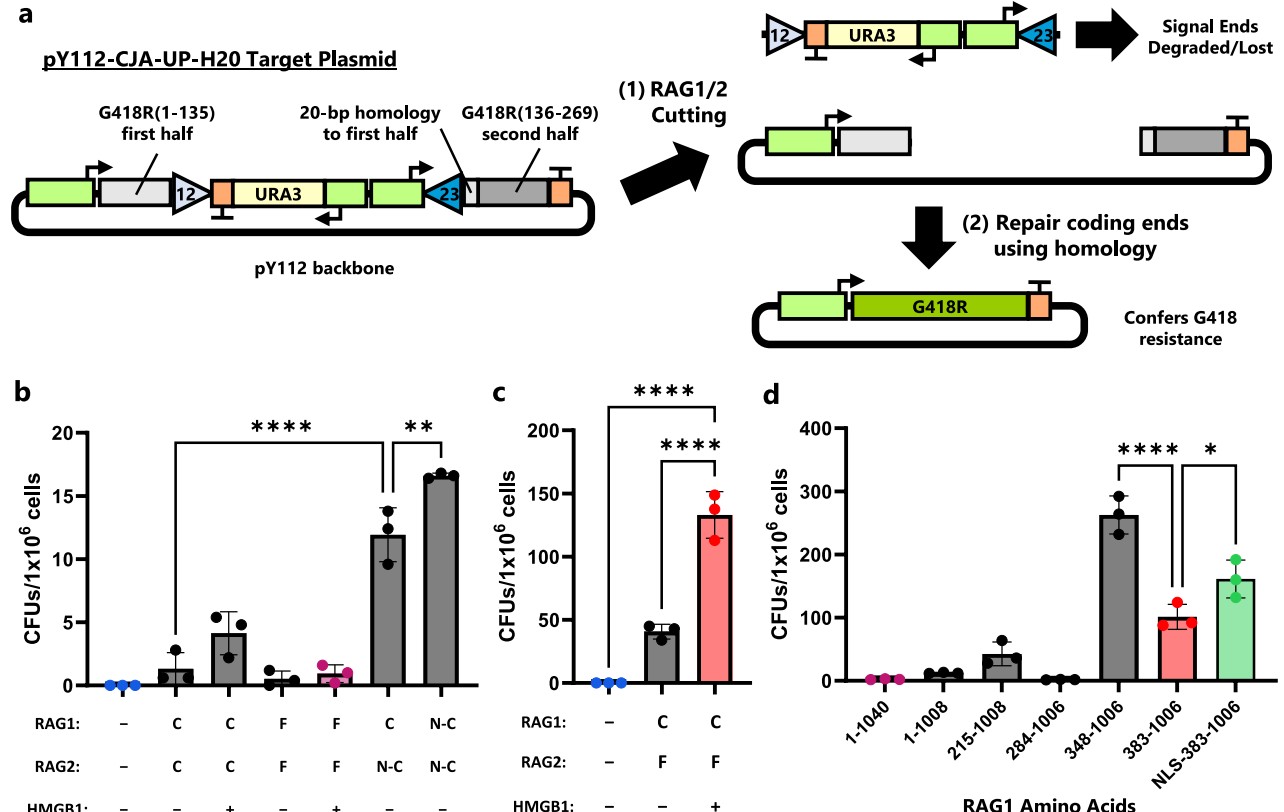

**Fig. 2 | Protein engineering increases activity of RAG-mediated, homology-assisted recombination of split antibiotic resistance gene. a** Diagram of pY112-CJA-UP-H20 recombination target plasmid. Triangles represent RSSs. Cutting by RAG1/2 leads to homology-assisted DNA repair of the *G418R* gene that confers resistance to the antibiotic G418. **b** Yeast strains were engineered with both RAG1 and RAG2 genes (either full-length, core, or core NLS variants) and optionally HMGB1. After 4-d induction in SG-Leu media, cell cultures were plated on Leu, G418 plates and colonies were counted after outgrowth. **c** *G418R* recombination with yeast strains assessing the combination of full-length RAG2 with RAG1core. Cells were plated after a 4-d induction. **d** *G418R* recombination with alternate truncations of RAG1 that include more regions of the full-length protein than the initial

RAG1core construct (which contains amino acids 383-1006). All strains include full-length RAG2 and HMGB1. Cells were plated after a 4-d induction. C = core protein, F = full-length protein, N-C = core protein with N-terminal nuclear localization signal, NLS = nuclear localization signal, CFU = colony forming unit. In (**b**–**d**) strains that appear in multiple plots are highlighted with a color, and data are presented as mean values ± SD; $n = 3$ biological replicates. Statistical significance was calculated with a one-way ANOVA and Tukey test (ns = not significant, $**p < 0.01$, and $****p < 0.0001$). From left to right, the highlighted *p* values are, in (**b**), $p = <0.0001$ and 0.0042; in (**c**), $p = <0.0001$ and $<0.0001$; and in (**d**), $p = <0.0001$ and 0.0148. Source data are provided in the Source Data file.

with the full-length RAG pairings might be attributable to poor expression of the full-length proteins in yeast, particularly RAG1. This would agree with numerous in vitro studies of V(D)J recombination which have normally relied upon the core protein sequences due to the difficulty of purifying the full-length proteins[38]. In this vein, utilizing yeast-codon optimized RAG1core and RAG2core sequences improved recombination by over 5-fold compared to wild-type mouse DNA sequences, once again with or without HMGB1 (Supplementary Fig. S4b). In addition, culturing yeast in glucose media, such that RAG1core and RAG2core expression were strongly repressed, yielded no G418-resistant colonies (Supplementary Fig. S4c).

Improvement in recombination rates mediated by co-expressing HMGB1 occurred in both full-length and core RAG contexts and were roughly similar (2- to 3-fold, Fig. 2b), which is supported by the context-independent mechanism of action for HMGB1—bending 23-RSS DNA to assist RAG-cleavage[19,39]. Therefore, HMGB1 was typically co-expressed in the following studies geared towards improving recombination levels. We next wanted to determine if the higher nuclear localization of NLS-RAG2core or full-length RAG2 impacted recombination rates compared to the cytoplasmic localized RAG2core. NLS-RAG2core led to an over 8-fold improvement in recombination compared to unaltered RAG2core, as expected based on its improvement in nuclear localization (Fig. 2b). Even better, pairing the full-

length RAG2 with RAG1core led to a 30-fold improvement relative to a strain with RAG2core (Fig. 2c).

**Varying RAG1 length further improves coding joint formation**

Based on the notable improvement full-length RAG2 provided compared to RAG2core, we reasoned that alternate truncations of RAG1 might also improve recombination. The RAG1core protein, spanning amino acids 383-1006, was designed to help purify catalytically active RAG1 for in vitro experiments[22,30,40]. However, multiple studies have shown that the removed regions of RAG1 are important for recombinase regulation or rate[41–43]. In particular, the RAG1 protein region between amino acids 348 and 383 contains a $C_2H_2$ zinc finger, while the region between 284 and 348 contains a zinc $C_3HC_4$ RING finger, which together comprise a dimerization domain[44,45]. In addition, the region between 215 and 284 contains multiple stretches of basic amino acids and is important for RAG1 localization and activity[31,41]. Therefore, we next tested if utilizing larger RAG1 variants affected recombination rates in our split-*G418R* assay by co-expressing RAG1 variants consisting of amino acids 348-1006, 284-1006, 215-1008, and 1-1008 with RAG2 and HMGB1, comparing to the RAG1core (383-1006) and full-length (1-1040) variants. Notably, RAG1(348-1006) had a 2.5-fold improvement relative to the RAG1core construct (Fig. 2d). In stark contrast, RAG1(284-1006) had very low activity, suggesting a complex

relationship between inclusion of distinct RAG1 domains and recombination activity in yeast. Lastly, adding an NLS to RAG1core provided a modest but clear benefit to recombination relative to RAG1core (Fig. 2d). Given its performance, this strain containing NLS-RAG1core, full-length RAG2, and HMGB1, and labeled AC518, was used for many characterization studies described hereafter.

To elucidate the effect RAG1 truncation has on expression, we tagged each truncation with eGFP and integrated them into yeast. Using flow cytometry, we measured the fluorescence of each construct (Supplementary Fig. S5a). In general, higher expression correlated with higher recombination activity (Supplementary Fig. S5b), but the correlation was not particularly strong, suggesting other factors such as localization or DNA-binding capability also impact recombination activity.

## RAG1 and RAG2 mutant characterization; additional protein factors influencing recombination

To confirm that the homology-assisted recombination was reliant on the catalytic activity of RAG1, we constructed two strains harboring RAG1 mutants with essential catalytic residues mutated, either D708N and E962Q[46]. While these catalytically dead mutants have been shown to have the ability to bind RSS sequences, recombination was completely abolished in these yeast strains, indicating RAG1 catalytic activity is required for homology-assisted V(D)J recombination (Supplementary Fig. S6a). Next, we tested if RAG1 nicking (without double-strand cleavage) was sufficient to stimulate recombination. RAG1 mutants R855A/R856A and R894A have been shown to be able to nick DNA adjacent to RSS sequences but lack the ability to cleave DNA via hairpin formation[47]. We created two strains that used these mutants, and RAG1 recombination was reduced over 500-fold for each strain (Supplementary Fig. S6a). Therefore, yeast homology-assisted recombination strongly benefits from a RAG1 that can efficiently cleave double-stranded DNA, similar to what is seen in mammalian cells.

We were also interested in employing engineered RAG2 variants that could enhance recombination. In particular, it has been shown in mammalian B cells that, in a cell-cycle dependent manner, RAG2 is phosphorylated at a threonine at residue 490 (T490) by CDK2 and degraded, limiting RAG2 activity to the G1 phase[48]. Yeast have a CDK2 homolog, CDC28, which may affect RAG2 in a similar manner. Mutation of T490 to an alanine has been shown to abolish this cell-cycle dependent degradation[48,49]. When we compared recombination rates mediated by full-length, wild-type RAG2 and a RAG2-T490A variant, RAG2-T490A did not mediate a significant improvement (Supplementary Fig. S6b).

One potential bottleneck for coding joint repair post-cleavage is the opening of coding end hairpins. In B cells, DNA-PKcs-phosphorylated-Artemis opens hairpins but a truncated Artemis (residues 1–413 of 692 total) has been shown to be constitutively active[50]. We integrated yeast-codon-optimized Artemis(1–413) into strain AC518 and tested the rate of recombination. Artemis(1–413) did not significantly increase the rate of recombination (Supplementary Fig. S6c). It is possible that another yeast protein is already opening the hairpins effectively as part of the homology-directed repair.

Encouraged by our improved recombination rates seen utilizing the mouse RAG1core and full-length RAG2 proteins, we tested the ability of the human RAG proteins to function in yeast, which had not been determined previously. A yeast strain expressing human RAG1-core, full-length RAG2, and HMGB1 had much lower recombination rates, <1% compared to its mouse counterpart (Supplementary Fig. S7a). While human RAG1core and RAG2core can be purified just like the more commonly used mouse proteins[51], we considered that the human proteins could suffer from poor expression in yeast. As with the mouse RAG1 truncations, we checked the expression of human RAG1core and RAG2 by fusing each to eGFP. Surprisingly, there was no

significant difference in fluorescence between mouse and human RAG1core or RAG2 (Supplementary Fig. S7b). To further validate the eGFP tag data, we performed a western blot to directly measure the amount of RAG2 produced by the cells. Human and mouse RAG2 were both easily detected at comparable levels (Supplementary Fig. S7c). Therefore, our data indicate that the poor performance of human genes relative to mouse is not caused by deficient expression. While our work marks the first time that human RAG sequences mediated recombination in yeast, we proceeded with the better performing mouse RAG proteins for further characterization and application.

## Non-protein factors influencing coding joint recombination activity in yeast

To verify that the mechanism of recombination in yeast required both RSSs, we generated variations of our pY112-CJA-UP-H20 detection plasmid that had mutations within the sequences of the 12-RSS, the 23-RSS, or both the 12- and 23-RSSs (Supplementary Table S4). Yeast strains harboring plasmids with mutated RSSs yielded no colonies. Furthermore, plasmids with two identical RSSs (12 + 12 and 23 + 23) also prevented recombination, showing that correct 12- and 23-RSSs are required and indicating that the traditional mechanism for RAG cleavage, including the 12/23 rule, was being followed in yeast (Fig. 3a). Next, in the split-G418R recombination assay, both RSSs are eliminated from the plasmid, such that recombination to form the coding joint-bearing plasmid is irreversible. Therefore, we were interested in seeing if the percentage of yeast harboring a recombined plasmid, and hence a G418R gene, in a culture would increase over time. We performed a time-coursed analysis of recombination levels in yeast cultures expressing RAG1core, full-length RAG2, and HMBG1, plating cells every two days for fourteen days total (Supplementary Fig. S8a). As might be expected, the percentage of yeast harboring recombined coding-joint plasmids increased over time, with a fivefold increase at fourteen days relative to four days.

In yeast, 20 bp of homology is approaching the lower limit needed to allow homology-mediated DNA repair, and was therefore chosen as our level of overlap in our homology-assisted coding joint formation plasmid, pY112-CJA-UP-H20[52]. We were curious if RAG1/2 expressing yeast strains could still mediate recombination if the level of overlap was reduced, so we tested the impact of utilizing plasmids with 0, 3, 6, 10, 15, 20, or 40 bp of homology between the two G418R fragments (Fig. 3b). As expected, the rate of recombination decreased drastically with decreasing overlap length, such that strains harboring plasmids with 10 bp of overlap had an extremely low level of recombination, while those with less had no detectable recombination. Sanger sequencing of four colonies confirmed that the repair was still accomplished without error with 10 bp of homology. In addition, we saw that wild-type yeast cells harboring split-G418R constructs with 40 bp of homology began to have detectable but still low recombination, suggesting that our utilization of the 20 bp overlap was optimal to detect recombination events driven by RAG1/2 activity.

To further increase the recombination rate, we reasoned that a high-copy, 2 μ plasmid might increase the likelihood of G418R formation relative to the pY112-CJA-UP-H20 construct which uses a low-copy, CEN/ARS origin of replication. After creating a 2 μ plasmid that is otherwise identical to pY112-CJA-UP-H20, however, we saw that the rate of colony formation was unchanged (Supplementary Fig. S8b), showing that plasmid origin of replication does not strongly impact recombination. To further characterize the impact of the genetic context, we integrated the coding joint target into the yeast genome. Recombination events were still easily detected, but the rate was much lower than what was seen for the original plasmid (Supplementary Fig. S8c).

To continue to gauge the importance of the recombination substrate plasmid design, we made additional alterations to test the importance of the split site within the G418R gene as well as the length

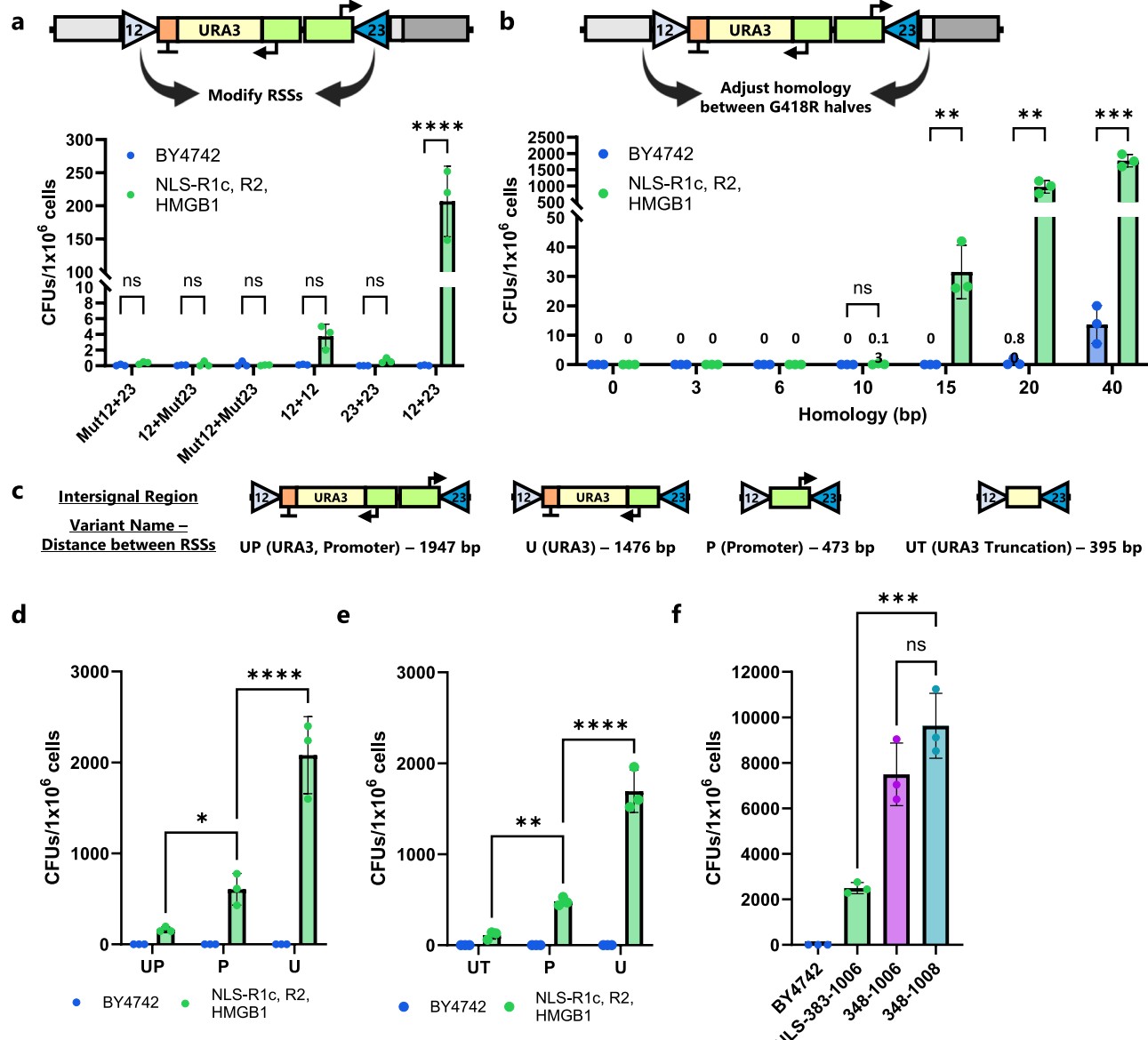

**Fig. 3 | Homology and intersignal sequence strongly influence *G418R* recombination assay. a** *G418R* recombination assay using substrates with mutated RSSs that cannot be recognized by the RAG proteins (Mut12 and Mut23) or nonstandard pairings (12 + 12 and 23 + 23). Cells were cultured for 4 d in SG-Leu prior to plating. **b** *G418R* recombination assay showing the effect of adjusting homology length between the two halves of the *G418R* gene in the recombination target plasmid. Cells were plated after an 8-d induction. **c** Diagram of the original intersignal region for "UP" plasmid design and the shorter variant sequences between the RSSs that were tested. For all alternatives, 20 bp of homology between the two halves of *G418R* was used. **d** *G418R* recombination assay varying the intersignal region between RSSs. See (**c**) for region descriptions. Cells were plated after a 4-d induction. **e** Similar to (**d**), *G418R* recombination assay testing additional intersignal regions between RSSs. Cells were plated after a 4-d induction. **f** *G418R* recombination assay using enhanced pYI12-CJA-U-H20 plasmid with

additional recombination strains. Recombination strains contained RAG2, HMGB1, and the specified variant of RAG1. Cells were plated after 4-d induction. In all figures, BY4742 are wild-type cells that do not contain V(D)J recombination genes. CFU = colony forming unit. In (**a**, **b**, **d**–**f**) data are presented as mean values ± SD; *n* = 3 biological replicates. In (**a**, **d**, and **e**), statistical significance was calculated using a two-way ANOVA and Tukey test. In (**b**), significance was calculated using multiple two-sided t-tests with a Holm-Sidak correction for multiple comparisons. In (**f**), statistical significance was calculated using a one-way ANOVA and Tukey test (ns = not significant, \*p < 0.05, \*\*p < 0.01, \*\*\*p < 0.001, \*\*\*\*p < 0.0001). From left to right, the highlighted *p* values are, in a, *p* = 0.9811, 0.9916, 0.9916, 0.7759, 0.9602, and <0.0001; in (**b**), *p* = 0.2606, 0.0078, 0.0028, and 0.0004; in (**d**) *p* = 0.0329, < 0.0001; in (**e**) *p* = 0.0017 and <0.0001; and in (**f**) *p* = 0.0001 and 0.1147. Source data are provided in the Source Data file.

and composition of the intervening DNA between the 12- and 23-RSS. Splitting the *G418R* gene in a different location (pYI12-CJA-UP-H20alt) did not substantially alter recombination rate, suggesting that the initial *G418R* split site was not exceptional (Supplementary Fig. S8d). Next, we created substrate plasmids in which we (a) removed the URA3 selection cassette but left the promoter pointing towards the 23-RSS, decreasing the distance between the RSSs from ~2 kb to ~0.5 kb (i.e., "P"), (b) removed the strong constitutive promoter but left the URA3

cassette, decreasing the distance between the RSSs to ~1.5 kb, ("U"), or (c) removed the promoter and reduced the URA3 cassette to a non-functional truncation of the *URA3* gene only ~0.4 kb in length ("UT", Fig. 3c). Removing only the larger URA3 cassette increased recombination activity by nearly 4-fold, while surprisingly, removing only the promoter led to an over 10-fold improvement in recombination (Fig. 3d). In contrast, the "UT" construct had the lowest activity of the three shortened intersignal sequences (Fig. 3e). These results highlight

**a**

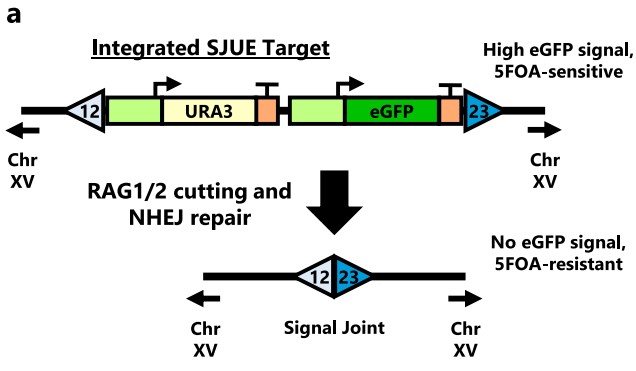

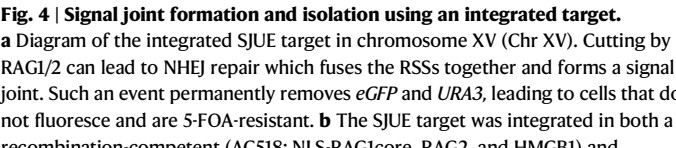

**b**

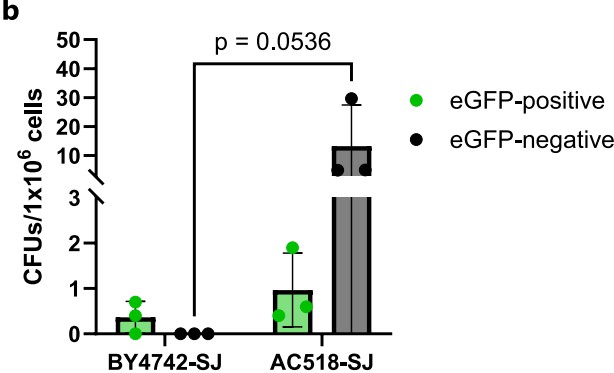

**Fig. 4 | Signal joint formation and isolation using an integrated target.**
**a** Diagram of the integrated SJUE target in chromosome XV (Chr XV). Cutting by RAG1/2 can lead to NHEJ repair which fuses the RSSs together and forms a signal joint. Such an event permanently removes *eGFP* and *URA3*, leading to cells that do not fluoresce and are 5-FOA-resistant. **b** The SJUE target was integrated in both a recombination-competent (AC518: NLS-RAG1core, RAG2, and HMGB1) and

recombination-incompetent (BY4742) strain. After an 8-d induction in YPG media, cells were plated on SD, 5-FOA plates. Both eGFP-positive and eGFP-negative colonies were separately counted. CFU = colony forming unit. In (**b**), data are presented as mean values ± SD; *n* = 3 biological replicates. Statistical significance was calculated using a two-way ANOVA with repeated measures across fluorescence types and Fisher LSD test. Source data are provided in the Source Data file.

the sensitivity that RAG proteins have to the region between the RSSs. They suggest that multiple factors, such as intersignal distance, transcriptional activity, or possibly the nucleotides adjacent to the RSS can combine to impact recombination efficiency.

We tested the best target plasmid with the highest-performing recombination strain we characterized earlier which uses RAG1(348-1006), RAG2, and HMGB1. RAG1core commonly ends at residue 1008, therefore we also built a similar strain which includes the additional two amino acids on the C-terminus (348-1008). Recombination rates reached nearly 10,000 CFUs/1×10⁶ cells—or 1% recombination—after four days (Fig. 3f). Unsurprisingly, we did not observe a significant difference between RAG1(348-1006) and RAG1(348-1008). The nearly 1% rate of RAG recombination is beginning to approach the rates seen using transfection in non-lymphoid, mammalian cell lines[16,41]. The rate was high enough to analyze events via flow cytometry and suggested that our yeast-based platform could be a viable alternative to mammalian cells for certain applications, such as studying defects in RAG1 and RAG2 proteins.

### RAG-expressing yeast strains can also create signal joints

To further demonstrate that our yeast recombination follows a mechanism similar to what takes place in B and T cells, we sought to detect signal joints. Signal joints are hallmarks of V(D)J recombination and are commonly used to assay recombination efficiency[40,42]. Stringent phenotypic assays are challenging to design for detecting signal joints because they are formed through the ligation of the two RSSs, making it impossible to recombine a split resistance marker as we have done for coding joint detection. To overcome this challenge, we designed a new construct, named SJUE, in which we flanked URA3 and eGFP expression cassettes with RSSs such that signal joint formation would remove both markers from the construct (Fig. 4a). Cells harboring a successful recombination event would be resistant to 5-fluoroorotic acid (5-FOA) supplementation and eGFP-negative. We included *eGFP* within the excised region to allow discrimination between cells that had excised *URA3* during signal joint formation and those with point mutations in the *URA3* gene, which can occur naturally. To ensure a consistent, single construct copy number, we integrated SJUE into the genome of strains BY4742 and AC518.

After an 8-day induction for each strain and then plating on 5-FOA plates, strain AC518 produced substantially more colonies than BY4742 (Fig. 4b). Importantly, for strain AC518, the majority of colonies were eGFP-negative, and while a BY4742 formed a small number of colonies, all were still fluorescent, indicating a signal joint had not

formed. In previous efforts to detect signal joints in yeast, they were created on high-copy plasmids and were detected via a nested PCR assay, so it is difficult to compare our rate of signal joint formation to rates of prior studies[28]. Importantly, our work marks the first time signal joint recombination events were isolated in a living yeast cell, and using Sanger sequencing, we were able to analyze the recombination sites directly from 15 colonies of 'signal joint positive' AC518. All were blunt ligations of the RSSs without error (Supplementary Fig. S9a). We also sequenced three colonies of AC518 that had maintained fluorescence, one of which had a *URA3* point mutation, as expected. Surprisingly, the other two colonies harbored signal joints that appeared to have formed using the 12-RSS and a cryptic 23-RSS located in the *URA3* terminator (Supplementary Fig. S9b). We did not detect evidence of signal joint formation in BY4742 cells. Our results appear to differ from previous work which showed a majority of recombination events using plasmid and integrated substrates underwent transposition[28]. However, unlike our design, their signal joints were not preserved in the genome after recombination, which may have biased their response toward transposition.

### Yeast-based analysis of RAG1 mutations deleterious toward V(D)J recombination activity

In mammals, deleterious RAG1 mutations have been shown to cause varying degrees of immunodeficiency, including T⁻B⁻ severe combined immunodeficiency (SCID) and Omenn Syndrome[53]. As many of these mutations directly affect the ability of RAG1 to bind or cut DNA, we hypothesized that recombination rates seen in our assay would inversely correlate with the severity of RAG1-mutation-associated immunodeficiency. To test our hypothesis, we selected five human RAG1 mutations known to have a range of clinical outcomes: T403P, G516A, R559S, R699Q, and K992E (Fig. 5a, c)[53]. We performed a mutant allele analysis by introducing each mutation into their matching position in mouse RAG1core (T400P, G513A, R556S, R696Q, and K989E) in the AC518 strain and then testing for altered recombination ability (Fig. 5b). The replaced mouse RAG1 amino acid was identical to the initial amino acid found in human RAG1 for each allele mutant, which is expected due to the >96% identity of the core proteins[20]. We then utilized the higher-activity pYI112-CJA-U-H20 target plasmid to assay for RAG-mediated recombination activity. Consistent with prior studies in a mouse B cell line[53], the T400P and R556S mutations essentially eliminated recombination (Fig. 5c). G513A and R696Q RAG1core mutant activity also correlated with their activity in mammalian cells, retaining 10% and 40% activity respectively relative to a

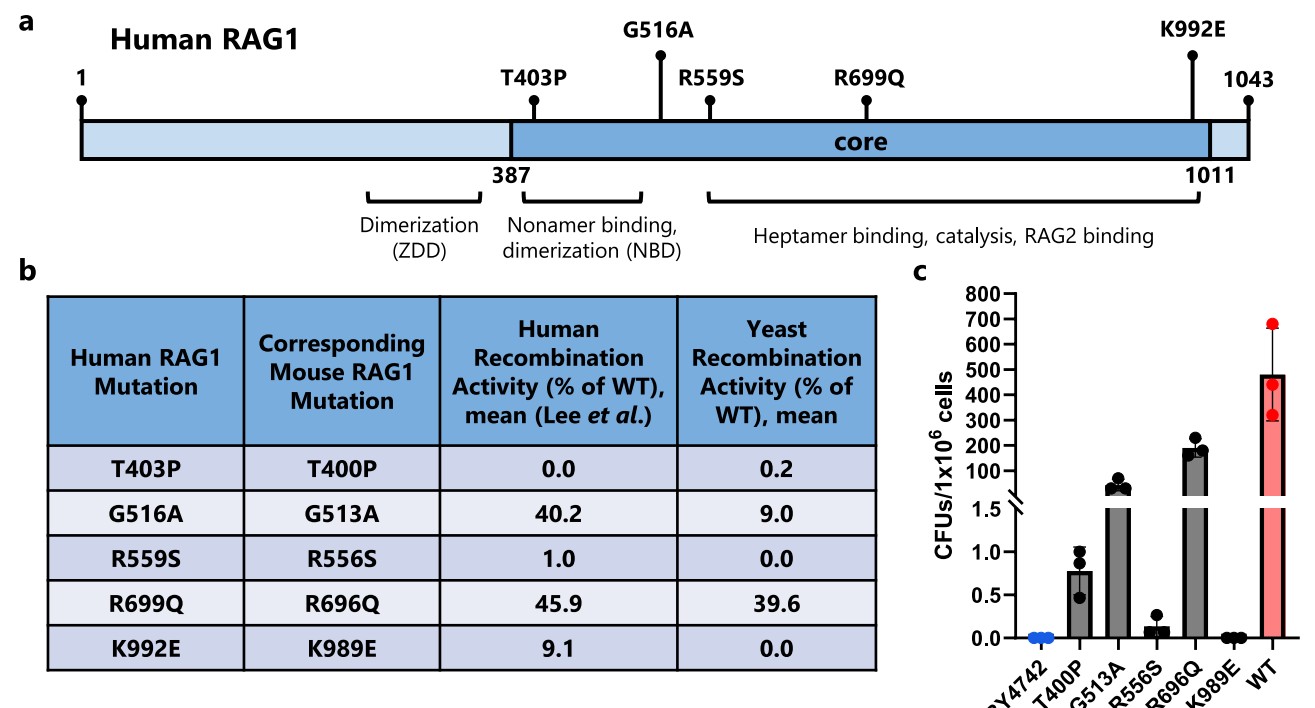

**Fig. 5 | Recombination efficiency of pathogenic RAG1 mutants in yeast correlates with activity in mammalian cells. a** Genetic map of human RAG1 showing the core region and the location of five mutations which have been shown to cause varying degrees of immunodeficiency. Certain domains of RAG1 and their function are highlighted, including the zinc dimerization domain (ZDD) and nonamer-binding domain (NBD). **b** Table of human RAG1 mutants and their corresponding mutation in mouse RAG1. Additionally, the previously reported activity of the mutants in mammalian cells is given[53] along with the activity measured in yeast in this work. WT = wild-type RAG1. **c** G418R recombination with pYI12-CJA-U-H2O plasmid. BY4742 are the base strain of yeast. All other strains contain mouse RAG1core with a mutation (or without in the case of wild type, WT), RAG2, and HMGB1. Cells were plated after a 4-d induction. CFU = colony forming unit. In (**c**), data are presented as mean values ± SD; n = 3 biological replicates. Source data are provided in the Source Data file.

strain with an unaltered RAG1core[53]. We did not detect any activity from K989E RAG1core, which represented a greater loss of activity than what we would expect based on studies in mammalian cells[53]. Still, coding joint formation in yeast largely correlates with RAG1 variant recombination in mammalian cells, demonstrating that our yeast strains might be used to predict pathogenic RAG1 mutant activity.

## Enabling in vivo, RAG-generated combinatorial diversity of fluorescent yeast

We reasoned that our engineered yeast could generate in vivo combinatorial diversity, an essential process towards immunoreceptor repertoire generation in humans. The ability to generate in vivo combinatorial diversity without the addition of exogenous DNA has not been demonstrated in an engineered model organism, but it could be a very useful tool for protein engineering[54]. Combinatorial diversity in V(D)J recombination arises because one of multiple V, D, or J gene segments (each with flanking RSSs) is randomly selected for recombination, such that the number of potential combinations is the product of how many possibilities exist for each gene segment.

We selected two model protein classes, fluorescent proteins and antibody fragments, to demonstrate and study the ability to create in vivo combinatorial diversity in our yeast-recombination strains. We first designed constructs that could be used to generate fluorescent protein combinatorial diversity, selecting three variants of *Aequorea victoria* GFP: eGFP (F64L, S65T), Sapphire (S72A, Y145F, T203I), and Azurite (F64L, Y66H, Y145F, V150I, V224R)[55,56]. Our combinatorial diversity substrates contained the first 62 amino acids of GFP, which are shared between eGFP, Sapphire, and Azurite, flanked by a 12-RSS, followed by DNA intersignal sequences and either two or three of the remaining gene sequence fragments of eGFP, Sapphire, or Azurite,

each flanked by a unique 23-RSS (Fig. 6a). These 3′ sequences had different codon usages to prevent native, homology-directed recombination. We reasoned that recombination events would be irreversible, as the 12-RSS is eliminated from the substrate plasmids following an event. In this way, a starting population of non-fluorescent cells induced to undergo homology-assisted, RAG-mediated recombination would create up to three new fluorescent subpopulations yeast cells, distinguishable via flow cytometry (Supplementary Fig. S10a–c).

Using a two-output combinatorial diversity plasmid substrate, in which a 23-RSS + eGFP 3′ fragment was followed by a 23-RSS + Sapphire 3′ fragment, we saw recombination that produced subpopulations of either fluorescent color, starting from a single non-fluorescent bulk population (Fig. 6b, c). We saw a higher prevalence of yeast cells expressing eGFP, the fragment closer in proximity to the 12-RSS. When the order of the eGFP and Sapphire fragments were switched, we saw a higher prevalence of Sapphire expressing cells, suggesting that shorter DNA sequences between 12- and 23-RSS enhanced recombination likelihood (Fig. 6c). A separate target plasmid with eGFP and Azurite fragments showed similar behavior to the initial eGFP-Sapphire construct (Fig. 6d). When utilizing a plasmid substrate in which yeast-based recombination could create sequences for all three colors, we were able to detect eGFP, Sapphire, and Azurite subpopulations after induction of RAG expression (Fig. 6e, Supplementary Fig. S10d). Again, the prevalence of each color's subpopulations decreased with increasing distance of the 23-RSS from the 12-RSS (prevalence eGFP > Sapphire > Azurite). In all these studies, yeast strains lacking RAG expression did not create any fluorescent populations, as expected. Thus, we have shown that RAG-mediated recombination of a single starting population can generate functional in vivo combinatorial DNA diversity and can be extended to multiple substrates.

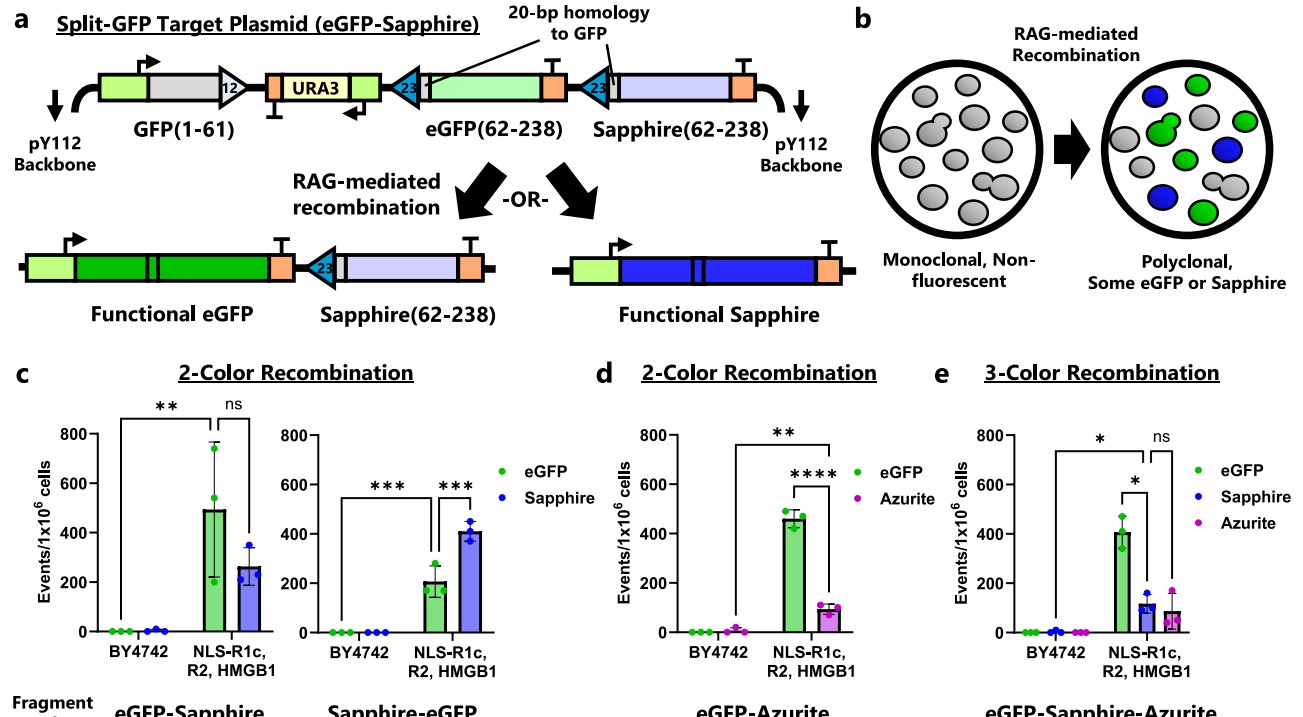

**Fig. 6 | RAG-mediated recombination of split-GFP construct generates diversity —up to three colors in vivo. a** Diagram of the pY112-CJCG-ES-H20 plasmid. The first portion of GFP is flanked by a 12-RSS followed by an intersignal region and then two modules each consisting of a 23-RSS, a gene fragment that creates eGFP or Sapphire, and a terminator. Triangles represent RSSs. Each gene fragment has 20 bp of homology to the first portion of GFP. Due to the 12/23 rule of RAG recombination, there are only two expected resolutions to recombination, one which creates eGFP and another which creates Sapphire. **b** Expected phenotype change for cells bearing the pY112-CJCG-ES-H20 plasmid as recombination is induced. Initially, all cells are non-fluorescent, but RAG-mediated recombination leads to subpopulations with eGFP or Sapphire fluorescence. **c** 2-color recombination of cells with target plasmids that have a fragment order of eGFP then Sapphire (pY112-CJCG-ES-

H20) or Sapphire then eGFP (pY112-CJCG-SE-H20). **d** 2-color recombination with fragment order eGFP then Azurite (pY112-CJCG-EA-H20). **e** 3-color recombination using a target plasmid with a fragment order of eGFP, Sapphire, then Azurite (pY112-CJCG-ESA-H20). For all tests, cells were cultured for 9-d in SG-Leu media prior to analysis with flow cytometry. BY4742 are wild-type cells that do not contain V(D)J recombination genes. In (**c–e**) data are presented as mean values ± SD; $n = 3$ biological replicates. Statistical significance was calculated with a two-way ANOVA with repeated measures across colors and Fisher LSD test (for **c** and **d**) or Tukey test (for **e**) (ns = not significant, $*p < 0.05$, $**p < 0.01$, $***p < 0.001$, $****p < 0.0001$). From left to right, the highlighted $p$ values are, in (**c**), $p = 0.0027$, 0.1741, 0.0001, and 0.0001; in (**d**), $p = 0.0012$ and $< 0.0001$; and in (**e**), $p = 0.0327$, 0.8738, and 0.0136. Source data are provided in the Source Data file.

## Extending in vivo combinatorial diversity generation to displayed scFvs

We next wanted to see if the ability to generate combinatorial diversity could be applied to antibody fragments, as 1) a key function of V(D)J recombination is making antibodies, and 2) yeast are an important platform for antibody sequence engineering and interrogation because of their ability display antibody fragments on their cell surface[8]. We engineered the *S. cerevisiae* surface display strain EBY100, to co-express NLS-RAG1core, full-length RAG2, and HMGB1, as in BY4742. We also expressed a new substrate plasmid in this recombination/display yeast strain, which was designed similarly to the combinatorial GFP substrate plasmids but utilized two antibody fragments (scFvs) that were derived from the same heavy chain-light chain ($V_H$-$V_L$) pairing (Fig. 7a). The two scFvs differed in the amino acid sequence of their $V_H$ complementarity determining region 3 (CDR3), such that one scFv binds prostate stem cell antigen (PSCA), while the other binds glypican3 (GPC3). Both PSCA and GPC3 are antigenic targets known to be overexpressed in certain cancers[57,58]. In our substrate plasmid, an AGA2 display protein fused to a DNA fragment of most of the scFv's $V_H$ is next to a 12-RSS, followed by an intersignal region and 23-RSSs preceding each 3′ fragment of the two scFvs. After inducing yeast cells harboring this scFv combinatorial diversity substrate, we indeed saw generation of yeast cell subpopulations displaying functional anti-PSCA and anti-GPC3 binding scFvs (Fig. 7b, Supplementary Fig. S11). In contrast, cells that did not have integrated RAG genes (BY4742) or cells

cultured in glucose were unable to generate scFvs. As with GFP recombination, we saw a higher rate of recombination at the 23-RSS most proximal to the 12-RSS, in this case generating more anti-PSCA than anti-GPC3 scFvs. While our demonstration here is far less complicated than bona fide V(D)J recombination, it does show that antibody combinatorial diversity can be created by RAG proteins in vivo and functionally characterized in yeast.

## Discussion

In this work, we engineered a yeast platform able to perform homology-assisted V(D)J recombination, utilized protein-based strategies to enhance recombination rates, and explored how non-protein (e.g., substrate plasmid) factors influence recombination. Recombination is dependent on the presence and localization of RAG1 and RAG2 and requires a functional 12-RSS and 23-RSS. By applying codon optimization, adding HMGB1, and using a truncated RAG1 and full-length RAG2, we achieved a homology-assisted coding joint recombination rate of nearly 1% after four days. Furthermore, using a separate assay, we successfully isolated cells that had generated signal joints in genomic DNA. Signal joints are not dependent on homology. Therefore, their isolation demonstrates the capability for yeast to repair DNA using the NHEJ pathway, albeit at lower levels than the homology-assisted coding joints. We then showed that our recombination platform could quantitatively predict deleterious mutations in RAG1, a common cause of various immunodeficiency disorders. Finally, we

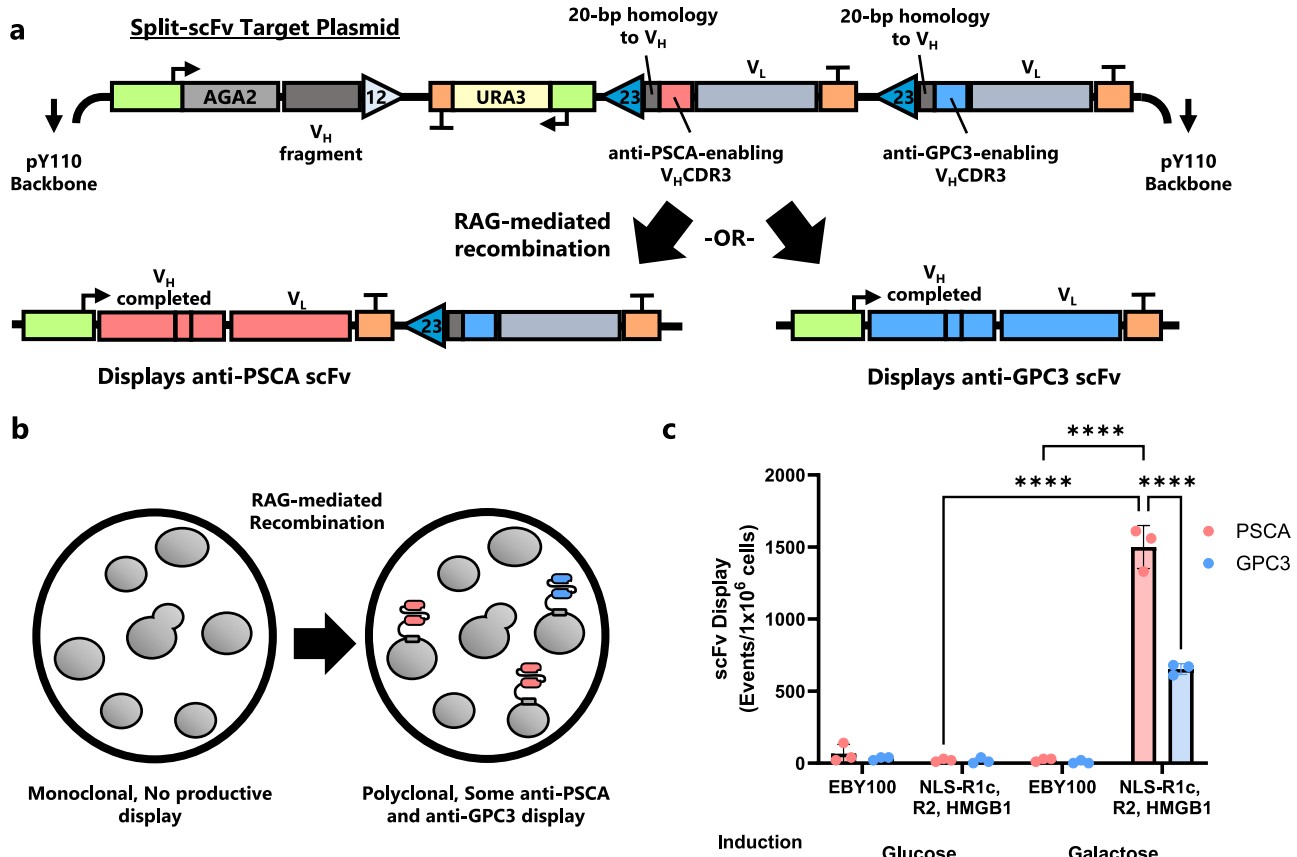

**Fig. 7 | Two functional scFv variants can be generated with recombination and displayed. a** Diagram of the pY110-CJCS-PG-H20 plasmid. AGA2 and a $V_H$ fragment are flanked by a 12-RSS followed by an intersignal region and then two modules each consisting of a 23-RSS, a gene fragment that contains a unique VHCDR3, and a terminator. Triangles represent RSSs. RAG-mediated recombination creates either an anti-PSCA or anti-GPC3 product which can be displayed on the surface of the cell. **b** Demonstration of expected phenotype change during RAG-mediated recombination of the split-scFv substrate. Initially, the cells cannot display a functional scFv. Depending on the outcome, a fraction of cells will display either an anti-PSCA or anti-GPC3 antibody after recombination. **c** scFv recombination results measured via flow cytometry. Wild-type EBY100 cells were compared to the recombination strain which expresses NLS-RAG1core, RAG2, and HMGB1. Cells were induced for 8 d in SG-Trp (Galactose) or SD-Trp (Glucose), then grown for 1 d in SD-Trp followed by 1 d in buffered SG-Trp. Due to fluorophore overlap, the cells were split into two separate staining reactions: one for PSCA and Myc and another for GPC3 and FLAG. In (**c**) data are presented as mean values ± SD; $n = 3$ biological replicates. Statistical significance was calculated with a two-way ANOVA with repeated measures to compare scFv display and Tukey test (****$p < 0.0001$). Source data are provided in the Source Data file.

used our platform to create the first instances of in vivo combinatorial diversity generation via recombination in yeast for two model protein classes: fluorescent proteins and antibodies.

In analyzing the impact of substrate sequences on recombination rates, we saw a complex interplay between the transcriptional activity and length of the intersignal region. While reducing the length between RSSs sometimes improved recombination ("U" and "P"), our shortest construct ("UT") did not follow this trend. In addition, we determined that, relative to our starting construct, removing the strong promoter in the intersignal region was beneficial for recombination in yeast. This promoter was originally included due to extensive work in mammalian cells showing the importance of RSS transcription in enabling V(D)J recombination[59]. A more comprehensive test with a large set of promoters of varying strength could be illuminating towards understanding the impact of intersignal transcriptional activity on recombination, and might yield a target with even higher recombination efficiency. Another possible explanation for the impact of the intersignal region is that recombination is highly sensitive to nucleotides near the nonamer of the RSS, but this has not been reported previously to our knowledge. Moreover, in the "UT" and "U" target plasmids, the two nucleotides nearest each RSS are identical, suggesting transcription and length as more influential variables. Importantly, by engineering multiple recombination targets, we were

able to develop selections that allowed for isolation of yeast cells harboring coding joints or signal joints, which had not been done previously in microbial cells. In addition to these standard products of V(D)J recombination, a wide variety of aberrant DNA repair can sometimes occur, such as transposition events[28,60]. In the future, it might be possible to catalog the occurrence of these events in yeast with different recombination substrates.

Yeast are well-established as a model eukaryotic cellular system that has been used to study several human disease states, including Huntington's disease, Alzheimer's disease, and cancer, through protein incorporation[61-63]. Indeed, a variety of strategies exist to humanize yeast in ways that help study human disease[64]. Here, we showed that yeast cells expressing RAG1 variants with pathogenic mutations demonstrated a decreased level of recombination rate that correlated with disease severity, expanding the use of yeast models towards immunodeficiencies. The RAG1 mutations tested should have a self-contained function, in that they are implicated directly in catalytic activity or DNA binding, and we saw a general agreement between our yeast system and prior results gauging RAG1 activity. Still, expression-related issues that arise in a heterologous host could impact these results.

To the best of our knowledge, our work is the first ever demonstration of a cell being engineered to generate in vivo combinatorial

protein diversity, which could be a useful tool to generate chimeric protein libraries. While our recombination is homology-driven, the RAG proteins have advantages over other site-specific nucleases which could conceivably be used to cleave DNA in vivo, such as CRISPR-Cas9 or meganucleases. With the RAG recombinase, both a 12-RSS and 23-RSS are required, and DSBs are simultaneously created exactly adjacent to two compatible gene segments. This increases the likelihood of productive recombination without increasing cell toxicity, and further removes the RSS recognition sites from recombined DNA. Because further recombination is prevented when all RSSs of a given type are excised, the potential for continuous DNA cleavage and repair until the 'smallest recombination option' remains is eliminated. Most importantly, as discussed below, future optimization work may eliminate the need to use any homology for RAG recombination. Thus, homology-directed repair can be seen as a steppingstone towards this end that we used to increase RAG expression and activity.

The ability to generate chimeric proteins in vivo unlocks many possible design strategies. For instance, by including more fragments with 23-RSSs (and better tuning the RSS function), a small library of scFvs could be generated, and additional diversity could then be induced by another system designed for targeted in vivo mutagenesis[65,66]. Because the recombination substrates are fully customizable, libraries could be generated that pair a light chain with various heavy chains, or vice versa, or fuse antibody gene segments together, such as, V gene segments with various DJ genes. Other classes of binders, such as designed ankyrin repeat proteins (DARPins) or nanobodies, could also be utilized in our system[67,68]. Future work might also employ multiple 12-RSSs and 23-RSSs to greatly increase the possible combinatorial diversity, as well as enabling two sequential recombination events, as occurs naturally during heavy chain recombination. Such demonstrations may be challenging given the relatively low rate of recombination. It is likely that a population could be enriched for coding joints prior to screening by using a negative selection, similar to the strategy we used to isolate signal joints. As recombination events increase with time, allowing the RAG proteins more time to recombine their target could also improve the recombination rate.

Although we were successful in generating combinatorial diversity, a weakness of our platform is that it does not create junctional diversity. The introduction of junctional diversity would most importantly require efficient production of coding joints via an error-prone repair pathway (i.e., not homology-assisted). Ideally diversity would be derived from both hairpin opening and non-templated polymerases, as occurs in vivo. We still lack direct evidence of hairpin formation and opening in yeast. While the results from the nick-only RAG1 mutants are consistent with the formation of a hairpin intermediate, adding constitutively Artemis to a strain with wild-type RAG genes did not improve the rate of recombination as might be expected with a hairpin. If a hairpin intermediate is formed, it remains unknown what factors are responsible for opening it during HDR. Certain yeast proteins have been shown to possess nuclease activity with hairpin substrates, such as SAE2/MRE11 or PSO2[69–71]. Hairpin opening in yeast may lead to end resection and commitment to an HDR pathway, which in turn would prevent error-prone, NHEJ-driven coding joint formation[72]. Similarly, it may be possible that NHEJ factors are being 'outcompeted' for coding-end binding by other, perhaps more prevalent, cellular proteins. In this vein, it may be important to bias DSB repair toward NHEJ through the deletion of various HDR genes[72].

A separate strategy to increase coding joint junctional diversity would be introduction of additional mammalian proteins, namely DNA-PKcs, Ku70, Ku80, and Artemis, that interact with the RAG complex and open hairpins in developing B cells, which may be more successful than sole introduction of constitutively-active Artemis[73,74]. While the act of hairpin-opening results in a certain level of junctional diversity in B cells, it would also be interesting to introduce the TdT non-templated DNA polymerase into our platform, as this enzyme might greatly amplify in vivo diversity[26]. Ultimately, engineering a yeast cell that undergoes V(D)J recombination as faithfully as possible would be an excellent tool to study mammalian development and disease and be a powerful platform for synthetic biology and protein engineering.

## Methods

### Strains, media, and general cloning techniques

NEB 10-beta *E. coli* (New England Biolabs) was used for all molecular cloning purposes. Most *S. cerevisiae* strains developed in this work were modifications of BY4742 (Ura-, Leu-). For experiments involving scFv recombination, a display-competent strain, EBY100 (Trp-, Leu-), was instead modified. *E. coli* were cultured in 5 mL of LB broth (Fisher Bioreagents) at 37 °C overnight with agitation. LB was supplemented with 34 μg/mL chloramphenicol (Sigma Aldrich) or 100 μg/mL ampicillin (Sigma Aldrich) antibiotic for selection. For selection of plasmids in *S. cerevisiae* with a LEU2 marker, synthetic leucine dropout media was prepared with either glucose (dextrose, SD-Leu) or galactose (SG-Leu), comprised of 0.69 g/L complete supplement mixture minus leucine (CSM-LEU, Sunrise Science), 6.7 g/L yeast nitrogen base (YNB) with ammonium sulfate (BD), and 20 g/L of glucose (Fisher Scientific) or galactose sugar (Sigma-Aldrich). Synthetic tryptophan dropout media (e.g., SD-Trp) or synthetic complete media (e.g., SD) were also prepared by replacing the CSM-LEU with 0.74 g/L CSM-TRP (Sunrise Science) or 0.79 g/L of CSM (Sunrise Science), respectively. G418 selection is incompatible with ammonium sulfate. Therefore, to generate SD-Leu + G418 plates, YNB was replaced with 1.7 g/L of YNB without ammonium sulfate (BD) and 1 g/L of monosodium glutamate (TCI) and supplemented with 300 μg/mL of G418 sulfate (Gold Biotechnology). For 5-fluoroorotic acid (5-FOA) selection, 1 mg/mL of 5-FOA (Gold Biotechnology) was supplemented to the media.

Rich media was also prepared by combining 10 g/L yeast extract (Thermofisher), 20 g/L peptone (Thermofisher), and 20 g/L glucose (YPD) or 20 g/L galactose (YPG). For integration selection plates, YPD was supplemented with 100 μg/mL nourseothricin (Gold Biotechnology) for *NATR* gene selection or 200 μg/mL hygromycin B (Invitrogen) for *HPH* gene selection. For yeast display of antibody fragments, SD-Trp or SG-Trp media was further buffered to pH 6.25 by adding 5.4 g/L $Na_2HPO_4$ and 8.56 g/L $NaH_2PO_4$•$H_2O$. For both yeast and *E. coli*, solid media plates were made with the addition of 20 g/L of agar (Fisher Scientific). For protein gels, 1x tris-glycine SDS (TGS) buffer was prepared by diluting 10x tris-glycine SDS buffer (Thermofisher) with ultrapure water. For western blot, 10x TBST buffer (Thermofisher) was diluted to a 1x concentration and was used for all membrane washes. When blocking or staining the membrane with antibody, 5 g of nonfat dairy milk powder was added in a final volume of 100 mL of TBST.

DNA was amplified by polymerase chain reaction (PCR) using KOD Hot Start polymerase (Sigma-Aldrich). Gibson or Golden Gate plasmid assemblies were performed following previously established protocols[65]. Many plasmids were made with Golden-Gate-compatible "parts" (i.e., promoters, genes, terminators, and backbones) that came from a plasmid collection previously generated for metabolic engineering applications[75]. *E. coli* transformations were carried out via electroporation. All plasmids were sequence confirmed with whole-plasmid, nanopore sequencing (Genewiz). For yeast transformation of plasmids as well as linear DNA fragments for genomic integration, a high-efficiency, lithium acetate transformation method was followed[76].

### Strain design and cloning

All *RAG1*, *RAG2*, and *HMGB1* genes and their variants expressed in this work were integrated into the yeast genome, generating a variety of engineered strains. A summary of every strain developed in this work is provided in Supplementary Table S1. In general, the strategy for all strain construction involved first making expression cassettes on integration-compatible plasmids. This was followed by co-

transformation of PCR-amplified, linear fragments of DNA, including a selection cassette, that would integrate simultaneously at a targeted locus. Each linear fragment had ~50 bp of homology to the adjacent fragments as described previously[75]. The two terminal fragments were amplified from genomic DNA and had ~400 bp of homology to the target locus as well as 50 bp of homology to the first and last linear fragment. Integration was directed toward sites which have been previously shown to support robust heterologous gene expression[77]. Following transformation, the presence of each integrated gene was confirmed through PCR of purified genomic DNA using a YeaStar Genomic DNA Kit (Zymo Research).

Two strains were created to easily visualize the nucleus and nucleolus during fluorescence microscopy. By tagging endogenous yeast proteins NAB2 and NOP56 with mCherry, it is possible to visualize the nucleus and nucleolus, respectively[33,34]. To build these strains, NAB2 and NOP56 along with their native promoter were PCR amplified from genomic DNA prepared from BY4742. Then, using Gibson assembly, two plasmids were made which fused NAB2 and NOP56 to mCherry on the C-terminus of each protein, connected by a short peptide linker. After amplification, the mCherry-tagged genes were integrated at YPRCΔ15 with noureseothricin (NAT) selection in BY4742 cells.

To prepare recombination genes for integration, first yeast-codon-optimized, full-length RAG1 and RAG2 (human and mouse sequences for each), and human HMGB1 were synthesized as linear fragments by Twist Bioscience. Sequences of synthesized genes can be found in Supplementary Table S2. Native mouse RAG1 and RAG2 sequences were isolated from CT26 mouse colorectal carcinoma cells using a mammalian genomic DNA prep kit (Quick-DNA Miniprep Kit, Zymo Research). Gene fragments were amplified and cloned into Golden-Gate-compatible vectors using Gibson cloning. In addition to the full-length RAG proteins, core versions of the proteins, RAG1core (amino acids 383-1006 of 1040 total) and RAG2core (amino acids 1-383 of 527 total), were also cloned. Identical truncations were made for the mouse wild-type genes, and a similar truncation was made for human RAG1 (amino acids 387-1011). The strong, galactose-inducible promoters of GAL1 and GAL2 were selected for RAG1 and RAG2 expression, respectively, while a strong, constitutive, heterologous promoter was selected for HMGB1. To augment subcellular localization, a nuclear localization signal (NLS) comprising the first 28 residues from histone H3 (HHT1) was selected for addition to RAG1core and RAG2core[35]. The NLS was amplified from genomic DNA and inserted at the 5′ end of both genes using Gibson assembly.

Integration-ready plasmids, which include promoters, terminators, and 60-bp flanks for integration were then cloned using Golden Gate. The full expression cassettes with flanking homology were then amplified from these plasmids along with a NAT selection cassette using PCR. These linear fragments were integrated simultaneously at the YPRCτ3 locus of strain BY4742 and plated on YPD + NAT. Due to the modular design, all desired combinations of RAG1, RAG2, and HMGB1 were created similarly.

To generate eGFP-tagged RAG plasmids, an eGFP expression plasmid was first made with Golden Gate from the part collection. Then fragments were amplified from the RAG expression plasmids and a fragment from the eGFP plasmid and combined in a Gibson assembly such that eGFP would be fused to the C-terminus of each protein with a (GGSGG)$_2$ linker. All eGFP-tagged RAG constructs were amplified and integrated into NAB2-mCherry and NOP56-mCherry strains at the YORWΔ22 locus with LEU2 selection.

Additional modifications were made to the original RAG expression plasmids to generate the deleterious RAG1 mutants, RAG2-T490A, and RAG1 truncation variants. For RAG mutants, primers were ordered to introduce the desired mutation, and the expression cassette was amplified in two pieces which were split over the mutation site. In this way, a subsequent Golden Gate reaction could reassemble the original construct with the new mutation. For the RAG1 truncation variants, DNA was amplified at the desired truncation position using the full-length RAG1 as template. Promoters and terminators were then added using Golden Gate assembly. All these protein variants were integrated at YPRCτ3 in BY4742 with NAT, similar to the standard constructs.

## Recombination target plasmid design and construction

A full list of the plasmids used to assay recombination is provided in Supplementary Table S3, and the set of target sequences is provided in a separate Supplementary Data file. For antibiotic resistance recombination target substrates, the plasmids were generated using iterative rounds of Golden Gate cloning. G418R, an aminoglycoside O-phosphotransferase from Aeromonas veronii which confers resistance to G418 (also known as Geneticin), was selected as a target resistance marker. First, plasmids pY112-psmTEF1-G418R-tEFM1 and pY112-pFBA1-klURA3-tklURA3 were made using Golden Gate and the part plasmid collection mentioned above, where klURA3 is from the yeast Kluyveromyces lactis and smTEF1 is from Saccharomyces mikitae. Next, plasmid pY112-CJA-UP-H20 was designed using PCR of these plasmids along with a strong, heterologous promoter, pspTDH3, followed by Golden Gate assembly. RSSs were added in this step with primers. Variants of pY112-CJA-UP-H20 were generated similarly using Golden Gate assembly. Backbone pY112 contains a LEU2 cassette and was selected for in yeast using −LEU media.

A separate target plasmid was designed to detect signal joints. URA3 and eGFP expression cassettes were made using Golden Gate. Then RSSs were added using nested PCR, and the final plasmid, pY112-SJUE, was assembled via Golden Gate. Due to plasmid incompatibility with 5-FOA selection, the SJUE target was amplified and integrated at the NRT1/GYP1 locus in strain AC518 using the URA selection built into the fragment.

To make constructs that would allow RAG-mediated production of combinatorial diversity within fluorescent proteins, plasmid pY112-psmTEF1-GFP-tEFM1 was first generated with Golden Gate. Gene segments for Sapphire and Azurite were synthesized by Twist Bioscience, each with a flanking 23-RSS and terminator. To avoid expansive homology between fluorescent protein coding regions, the Sapphire (S) and Azurite (A) gene sequences were codon optimized to remove homology regions greater than 10 bp to each other and to eGFP (E). From these components, plasmids pY112-CJCG-XX-H20 (where XX = ES, SE, EA, and ESA) were assembled by Golden Gate of fragments generated using PCR with tailored primers. The region between the RSSs was taken from pY112-CJA-U-H20. Single color controls for eGFP, Sapphire, and Azurite were also generated that had intact genes and the smTEF1 promoter. These plasmids were made by DNA amplification using modified primers and the recombination plasmids as template followed and Golden Gate.

Constructs that would allow RAG-mediated production of combinatorial scFv diversity were designed in a similar manner to the split-GFP plasmids. scFvs sequences were taken from a CDR3 library previously sorted against both PSCA and GPC3 followed by isolation. AGA2 and the first half of the scFv were followed by a 12-RSS and an intersignal region. Then a 23-RSS, second half of the PSCA scFv, and a terminator were followed by a second 23-RSS, the second half of the GPC3 scFv, and another terminator. The C-terminus of each scFv fragment was tagged with a unique peptide (Myc or FLAG), making it easier to discriminate which fragment recombined with the first half. The CDR3 which confers GPC3 targeting, along with a flanking 23-RSS and terminator, were synthesized by Twist Bioscience to remove homology to the anti-PSCA sequence. This plasmid, pY110-CJCS-PG, was transformed into a strain capable of both display and recombination. Plasmid pY110-CJCS-PG was created with Golden Gate using tailored primers. Backbone pY110 contains a TRP1 cassette and was selected for in yeast using −TRP media.

## GFP-tagged yeast expression, yeast microscopy, and image analysis

To quantify the relative expression of eGFP-tagged RAG1 and RAG2, cells were grown overnight in 0.5 mL of YPG media. $1 \times 10^6$ cells were collected and rinsed with PBS. eGFP fluorescence was then measured using a BD FACSMelody or Beckman Coulter Cytoflex. Data were analyzed and median fluorescence values gathered using FlowJo software. For confocal microscopy, yeast strains were grown overnight in 0.5 mL of YPG media. In the morning, yeast cultures were diluted to a concentration of $0.5 \times 10^7$ cells/mL in 0.5 mL of fresh, warm YPG, and then cells were allowed to grow for an additional 6 h. $1 \times 10^7$ cells were collected, rinsed with phosphate buffered saline (PBS), then resuspended in 10 uL of PBS and transferred to a glass slide and protected with a No. 1.5 glass coverslip. Z-stack images with a spacing of 0.5 μm were collected using a Zeiss LSM 700 microscope with a 63x objective lens using oil immersion. Instrument gain was held constant for all measurements except for the eGFP channel of RAG1 and RAG1core samples which have much dimmer eGFP signals.

Images were analyzed using Fiji/ImageJ to quantify the colocalization of the eGFP-tagged RAG proteins with mCherry-tagged NAB2 or NOP56. First, a maximum intensity z-projection was applied to the eGFP and mCherry channels for each image. A region of interest was created for each image by modifying the brightfield channel using the threshold and binary operations in ImageJ to create a mask for where cells were located. Then, the Pearson coefficient (no threshold) of each image was calculated using the Coloc 2 plugin in ImageJ. Each eGFP-tagged construct was imaged in biological triplicate, which were combined to calculate the mean and standard deviation.

## RAG2 western blot

To extract protein for western blot, cells were grown overnight in 10 mL YPG to a density $2.5 \times 10^7$ cells/mL. At this point, 1 mL of culture was collected and spun down and the supernatant was removed. Next, the cells were resuspended in 800 μL of 0.1 M NaOH and allowed to incubate at room temperature for five minutes. The cells were then spun down again, and the supernatant was completely removed. The pellets were resuspended in 60 μL of reducing Laemmli SDS sample buffer (Thermofisher) and incubated at 95 °C for five minutes. After one final spin down, the supernatant containing the soluble proteins was transferred to a new tube.

To perform the western blot, 6 μL of isolated protein solution were loaded per well into a 4-20% Novex tris-glycine gel (Thermofisher) and then run at 170 V for 1 h while submerged in TGS buffer. The proteins were then transferred from the gel to a PVDF membrane using a Power Blotter Select Transfer Stack with the Power Blotter instrument. The membrane was washed and blocked with 40 mL TBST buffer containing 5% nonfat milk and gently mixed for 1 h on a rocker at room temperature. Next, the membrane was stained with primary, polyclonal, rabbit anti-RAG2 antibody (Thermofisher) at a 1:1000 dilution in 25 mL TBST with 5% nonfat milk for one hour. The membrane was washed with 1x TBST to remove excess primary antibody. The cells were then stained with a secondary, polyclonal, goat anti-rabbit-IgG antibody tagged with horse radish peroxidase (HRP, Thermofisher) at a 1:2500 dilution. The membrane was washed one final time with TBST. The HRP signal was then detected by adding 5 mL of SuperSignal™ West Pico PLUS chemiluminescent substrate (Thermofisher). The membrane image was then recorded using a GE Amersham Imager 600. To compare the amount of protein loaded for each sample, another tris-glycine gel was prepared exactly as described above and then stained with Coomassie Brilliant Blue (Thermofisher).

## G418 resistance recombination assay

Prior to growing in liquid culture, cells were streaked on SD-Leu plates and allowed to grow for 2 days. Single colonies were picked from fresh plates and grown in 0.5 mL of SG-Leu in a round, 48-deepwell plate at 30 °C using a microplate shaker set to 700 rpm. After culture, the optical density (OD) was measured using a spectrophotometer from which cell concentration was calculated. Every two days, cultures were reinoculated into fresh SG-Leu media at a concentration of $0.2 \times 10^7$ cells/mL. At indicated times points, yeast cells were plated on 6-cm diameter SD-Leu + G418 plates and spread with beads. To aid counting, the total amount of plated cells was varied based on the recombination efficiency of the strain. If the plated volume was less than 100 μL, sterile PBS was added before the culture such that the combined volume was 100 μL. Cells were allowed to grow for 2–3 days prior to manual colony counting. For sequence analysis of recombined colonies, colony PCR followed by Sanger sequencing (Genewiz) was carried out.

## 5-FOA signal joint detection assay

A slightly modified protocol to that of the G418 resistance test was followed for signal joint detection. YPG media was used in place of SG-Leu, and cells were passaged every day rather than every 2 days. After 8 days, cultures were plated on SD, 5-FOA plates, and cells were allowed to grow for 2 days.

## Combinatorial GFP recombination assay

Beginning with freshly plated single colonies, yeast were cultured in 0.5 mL of SG-Leu in a round, 48-deepwell plate at 30 °C using a microplate shaker set to 700 rpm. Every two days, yeast cultures were reinoculated into fresh SG-Leu media at a concentration of $0.2 \times 10^7$ cells/mL. Cells were analyzed on a BD FACSMelody flow cytometer. Data were analyzed and recombination events were counted using FlowJo software with gates determined according to single color controls.

## Combinatorial scFv recombination assay

Again, starting from freshly plated single colonies, yeast cells were cultured for 8 days in 0.5 mL of SG-Trp or SD-Trp, with reinoculation into fresh media at a concentration of $0.2 \times 10^7$ cells/mL every two days. Yeast cultures were then reinoculated at a concentration of $0.5 \times 10^7$ cells/mL in buffered SG-Trp and grown overnight to induce scFv surface display. To discriminate antigen binding to the two substrates, each sample was treated with two separate stains, one to detect PSCA binding and the other to detect GPC3 binding. $2 \times 10^6$ cells were collected and rinsed with PBSF (PBS supplemented with 1 mg/mL bovine serum albumin), then stained with either biotinylated PSCA (ACROBiosystems) at 2 μg/mL or biotinylated GPC3 (ACROBiosystems) at 6 μg/mL for 60 min. After rinsing with PBSF, cells were stained with 10 μg/mL Streptavidin-PE (Thermofisher), and either 0.2 μg/mL anti-Myc-AlexaFluor647 (Cell Signaling Technologies, for cells that had been stained with PSCA) or 1.6 μg/mL anti-FLAG-APC (BioLegend, for cells that had been stained with GPC3). After 30 min, cells were rinsed again, and then resuspended in 400 μL PBSF. Fluorescence measurements were then collected for each sample using a BD FACSMelody. Due to the large difference between APC and AlexaFluor647 fluorescence intensity, different gain values were used for the PSCA/anti-Myc-stained cells versus the GPC3/anti-FLAG-stained cells. Data were analyzed and double positive events (with both antigen binding and peptide tag presentation) were counted using FlowJo software. Gates were determined using controls that display either the PSCA or GPC3 scFv.

## Statistics

All experiments were performed in biological triplicate ($n = 3$), except where noted, and charts show the mean with error bars representing ± one standard deviation (SD). All statistical tests were performed with Graphpad Prism software.

**Reporting summary**

Further information on research design is available in the Nature Portfolio Reporting Summary linked to this article.

## Data availability

Annotated target plasmid sequence files are included in the Supplementary Data file. Source data for all bar charts along with original gel and western blot images are available in the Source Data file. Source data are provided with this paper.

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

## Acknowledgements

We thank Eric M. Young (Department of Chemical Engineering at Worcester Polytechnic Institute) for the kind gift of the modular plasmid set for tunable yeast gene expression and Francesca Storici (School of Biological Sciences at Georgia Institute of Technology) for providing strain BY4742. J.B. acknowledges funding support from the Beckman Young Investigator Award from the Arnold and Mabel Beckman Foundation and from the NIH (Award Number 1DP2CA280622-01). The content is solely the responsibility of the authors and does not necessarily

reflect the official views of the National Institutes of Health or the Beckman Foundation.

## Author contributions

Conceptualization: A.P.C. and J.B.; Experimental investigation: A.P.C., J.S., S.Y., L.S.C., C.Y., O.M.I., H.A., and S.D.; Data processing and visualization: A.P.C., J.S.; Manuscript writing and editing: A.P.C. and J.B.; Funding Acquisition: J.B.

## Competing interests

The Authors declare the following competing interests: J.B. and A.P.C. have filed a patent related to this work. The other authors declare no competing interests.
