## [Transparent Peer Review file · Nature Communications]

Generating combinatorial diversity via engineered V(D)J-like recombination in *Saccharomyces cerevisiae*

Corresponding Author: Dr John Blazeck

Version 0:

Reviewer comments:

Reviewer #1

(Remarks to the Author)

The motivation of this paper is that creating a “yeast strain that could diversify antibody sequences in a manner similar to B cells could be a helpful tool both towards studying aspects of V(D)J recombination and towards generating combinatorial diversity *in vivo*”. Indeed, this study takes the very initial steps towards both basic and applied scientific goals. The authors use extensive synthetic biology and protein engineering tricks were necessary for the limited recombination demonstrated by yeast in this paper. These include protein truncation, alterations in spacer and homology regions of DNA, incorporation of different NLS tags. They found that the combination of a 20 bp homology arms between recombined cassettes, murine NLS-RAG1 core, murine full-length RAG2, and human HMGB1 could yield on-target recombination from approx. 0.02%-0.2% in yeast. The authors further demonstrate that a signal joint was isolated using a joint eGFP/URA3 assay, and with appropriate negative controls. The authors demonstrate this system in three ways. First, they show a correlative analysis between RAG1 mutants identified in humans and in the yeast measurements. Second and third, they demonstrate recombination between different fluorescent proteins (3 unique sequences) and antibody fragments (2 unique sequences). As a suitable starting point for the overall goals, this paper is sufficient. In my opinion, no further experiments are required as the rigor demonstrated is quite high. Even though limited conclusions can be found in individual experiments (for example, the impact of spacer length and promoter on recombination efficiency), the sheer number of experiments performed give a trove of information for the next papers seeking to optimize this system. I have some minor comments on framing in part, and perhaps on the likelihood that a strain capable of high efficiency recombination can be constructed.

1. It would help the manuscript if the end goal would be described in the introduction and referred to in the discussion. Since the efficiency of recombination in mammals for this system I think is quite high (recombination can occur in most cells given the appropriate signals), is the goal to recapitulate mammalian efficiencies? Would the end applications require a less efficient process, and if so, what are these approximate numbers? Is it 10%? 1%?

2. The abstract performs a metabolic engineering trick of reporting a fold amount (500-fold). This seems quite impressive until you note in the text that you are starting with essentially no recombination. I'd request that the authors report best recombination efficiencies in the abstract.

3. I may have missed this in the discussion, but I didn't see the impact of temperature or of the functional reconstitution of these murine and human components on potential causes for low efficiencies. More generally, I also didn't see comparison with *in vitro* reconstituted recombination. While outside of my field, I think the biochemistry and *in vitro* reconstitution of this system is mature and well worked out, and this literature may provide additional clues.

4. The abstract statement below is overly broad, and ideally should be excised or otherwise constrained. What counts as a demonstration of *in vivo* generation of combinatorial genetic diversity? Wouldn't formation of functional antiviral RNA proteins using CRISPR arrays qualify? What about homologous recombination of different gene fragments in fungi and yeast?

“...which we believe to be the first ever demonstration of *in vivo* generation of combinatorial genetic diversity in a non-vertebrate cell.”

5. The inclusion of Figure 1 first is a bit confusing. If the point of figure 1 and 2 is the ultimate development of strain AC518, I would recommend combining the essential bits of figure 1 & 2 and porting the remainder to the SI. I also think adding sequences that are currently in the SI may improve the figures, but I'll defer to the authors here.

Reviewer #2

(Remarks to the Author)

Manuscript ID: NCOMMS-24-81016

Title: Generating combinatorial diversity via engineered V(D)J-like recombination in *Saccharomyces cerevisiae*"

Authors: Cazier et al.

Cazier et al. report developing a yeast-based homology-driven V(D)J-like recombination system using engineered substrates and expressed RAG1 and RAG2 proteins. This idea of establishing ectopic RAG-mediated V(D)J recombination in yeast is not novel, which the authors acknowledge, but they do advance the reported efficiency of the recombination events by incorporating some clever ideas to optimize RAG expression and localization, incorporating homology-assisted repair, and also by including HMGB1/2 proteins. The system is highlighted for its potential utility in generating chimeric protein libraries. After establishing the baseline recombination activity with murine RAGs, they go on to evaluate how the system responds to human RAGs and mutant RAG proteins, as well as substrate mutations of various types (e.g. RSS mutations, intersignal distance, and combinatorial rearrangement). While there are some nice advances to the yeast RAG recombination system, there are several aspects of this work that make it more incremental than groundbreaking in its present form. These concerns are detailed below.

Major concerns:

1. A strength of the work is that the rearrangement events are pretty clearly RAG- and RSS-dependent. However, coding joint formation is not observed without homology-directed repair. The authors imply that because they observe signal joints, the reaction must involve classical introduction of double-strand breaks and NHEJ – the hallmarks of V(D)J recombination. However, there are no experiments to demonstrate that DNA breaks are actually generated via the known RAG-mediated nick-hairpin mechanism. Since the RAG proteins can nick DNA, the rearrangement events may instead be progressing through sequential nicking without hairpin formation being an intermediate. Elegant assays have been done in the V(D)J recombination field to establish the cleavage mechanism, so similar experiments should be feasible. Furthermore, while the data establish consensus RSSs are required for recombination, the authors do not actually test whether the rearrangement follows the 12/23 rule in Fig. 3, which is another hallmark of V(D)J recombination. To show that, one would have to compare rearrangement frequencies between 12/23 and 12/12 or 23/23 recombination substrates. These are the major reasons that the work falls short of representing a significant advance to the field.
2. Another limitation is the lack of testing to see if hairpin nucleases could augment rearrangement efficiency, presuming that the coding ends are indeed sealed as DNA hairpin structures. While there are no direct homologs to Artemis and DNA-PK, it would be good to at least explore this question a bit more. Have the authors ever tried expressing constitutively active Artemis to see if that improves recombination activity? An activating truncation mutant has been described in the literature.
3. Expression levels of the RAG proteins were never tested. Hence, it is unclear whether differences in expression can fully explain differences in recombination activity.

Minor issues:

1. The authors report human RAGs perform poorly in the recombination assay, but it is unclear why that might be. Here again, expression data would be helpful in interpreting this result.

Version 1:

Reviewer comments:

Reviewer #1

(Remarks to the Author)

The authors have satisfactorily addressed the points raised in the initial review.

Reviewer #2

(Remarks to the Author)

Re: NCOMMS-24-81016A:

The authors have performed several new experiments to address prior reviewer concerns. These include testing RAG1 "nick-only" mutants to indirectly show RAGs cut DNA in yeast through a "nick-hairpin" mechanism, testing recombination substrates containing 12/12 and 23/23 RSS pairs to show V(D)J recombination in yeast follows the "12/23 rule", and testing a constitutively active form of Artemis to determine whether recombination activity is stimulated by potential hairpin opening (no effect was seen). The authors also included experiments designed to evaluate RAG protein levels by immunoblotting. The addition of these experiments have strengthened the conclusions. Nevertheless, given the inefficiency of the recombination, and since there is no direct evidence that the "nick-only" mutants are actually only nicking DNA in yeast *in vivo*, the mechanism of cleavage remains indirectly tested. The authors should at least include a caveat stating this limitation in the discussion, and acknowledge that at present, hairpin opening, and hence the activity of Artemis, can also not be directly confirmed.

In the rebuttal letter, in response to Reviewer #1, the author makes some surprising statements regarding the state of the field regarding RAG biochemistry. The first is to suggest that full-length RAG1 has not been purified. However, Swanson's group in particular has purified full-length RAG1 and characterized its activity *in vitro* (e.g. PMIDs 14612436, 19233873, 22157821). Their finding in PMID 19233873 that full-length RAGs were slightly less active than core RAGs *in vitro* is in line with the authors findings. The second was the statement that "in vitro techniques can only be used to examine DNA nicking

or cleavage but not the joining events which are the usual in vivo readout". However, the Gellert group has reported on cell-free V(D)J joining (PMID: 9242409). The authors should also be aware that the "nick-only" mutants are known to stimulate homologous recombination (PMID: 15084256). In that regard, it is noteworthy that these mutants have distinct activities in yeast, with R855A/R856A mutant showing no recombination, and the R894A mutant showing partial activity. The latter observation could in fact support the idea that RAG nicking, rather than RAG-mediated hairpin formation, is what is initiating recombination in this system. Since the relative rate of nicking between WT and R894A in this system is unknown, the fraction of recombination attributed to RAG nicking could be quite high. This is an additional reason why a caveat statement is recommended.

Generating combinatorial diversity via engineered V(D)J-like recombination in *Saccharomyces cerevisiae*

Andrew P. Cazier, Jaewoo Son, Sreenivas Yellayi, Lizmarie S. Chavez, Caden Young, Olivia M. Irvin, Hannah Abraham, Saachi Dalvi, and John Blazek

Response to Reviewers

We thank the reviewers for their valuable comments and requests. We believe we have addressed all the reviewers' concerns, and importantly the changes have strengthened our conclusions and improved the manuscript. Below, we give our detailed response to each of the reviewers' comments.

RED – Reviewer comment BLACK – Author response

Reviewer #1 (Remarks to the Author):

The motivation of this paper is that creating a “yeast strain that could diversify antibody sequences in a manner similar to B cells could be a helpful tool both towards studying aspects of V(D)J recombination and towards generating combinatorial diversity in vivo”. Indeed, this study takes the very initial steps towards both basic and applied scientific goals. The authors use extensive synthetic biology and protein engineering tricks were necessary for the limited recombination demonstrated by yeast in this paper. These include protein truncation, alterations in spacer and homology regions of DNA, incorporation of different NLS tags. They found that the combination of a 20 bp homology arms between recombined cassettes, murine NLS-RAG1core, murine full-length RAG2, and human HMGB1 could yield on-target recombination from approx. 0.02%-0.2% in yeast. The authors further demonstrate that a signal joint was isolated using a joint eGFP/URA3 assay, and with appropriate negative controls. The authors demonstrate this system in three ways. First, they show a correlative analysis between RAG1 mutants identified in humans and in the yeast measurements. Second and third, they demonstrate recombination between different fluorescent proteins (3 unique sequences) and antibody fragments (2 unique sequences). As a suitable starting point for the overall goals, this paper is sufficient. In my opinion, no further experiments are required as the rigor demonstrated is quite high. Even though limited conclusions can be found individual experiments (for example, the impact of spacer length and promoter on recombination efficiency), the sheer number of experiments performed give a trove of information for the next papers seeking to optimize this system. I have some minor comments on framing in part, and perhaps on the likelihood that a strain capable of high efficiency recombination can be constructed.

1. It would help the manuscript if the end goal would be described in the introduction and referred to in the discussion. Since the efficiency of recombination in mammals for this system I think is quite high (recombination can occur in most cells given the appropriate signals), is the goal to recapitulate mammalian efficiencies? Would the end applications require a less efficient process, and if so, what are these approximate numbers? Is it 10%? 1%?

We thank the reviewer for highlighting this important question regarding our goals for recombination efficiency. In this manuscript, our goals for the yeast V(D)J recombination platform are, as you note above, to design a “helpful tool both towards studying aspects of V(D)J recombination and towards generating combinatorial diversity *in vivo*.” Ultimately, both these goals will benefit from high recombination efficiencies. However, as we demonstrate in this paper, far lower efficiencies (i.e., in the range of 0.02% to 1% recombination) can still be used to generate limited diversity and study RAG1 mutants. Improving recombination rate was particularly helpful towards allowing flow cytometry analysis, which is burdensome with recombination rates below ~0.01%.

We note that while V(D)J recombination is likely very efficient inside the body, the exact rate is not well quantified. It may be most analogous to compare our recombination rates to those seen using transfection of RAG genes in immortalized, non-lymphoid mammalian cells targeting extrachromosomal RSSs. When using transfected RAG1 and RAG2 expression vectors on target plasmids, mammalian systems yield relatively low recombination rates, e.g., around 0.1-1% signal joint formation (reference McMahan *et al.*, 1997). In general, the highest recombination efficiencies we have seen in yeast are roughly 1% (**Figure 3f**, see our response to the second comment below). Thus, our homology-assisted recombination in yeast may already approach mammalian cell recombination efficiencies in some respects, and it will be worthwhile to compare future iterations of our platform to information about recombination efficiency in individual B or T cells (which we would expect to be higher but likely not at 100%) as that data potentially becomes available.

We also note that yeast have multiple techniques described to perform a negative selection, i.e., using 5-FOA to select for the loss of the URA3 gene, which we leveraged to isolate signal joints in Figure 4. Using a negative selection could also mimic what occurs during B and T cell maturation *in vivo* where cells that do not produce a functional receptor are not stimulated for expansion and ultimately die. Lastly, we have shown that homology-assisted coding joints increase with time. Therefore, simply inducing the RAG proteins for more time should increase the recombination rate as well. In future work, these strategies could potentially be applied to enable effective screening of combinatorial libraries.

The following was added at line 595:

“The nearly 1% rate of recombination is beginning to approach the rates seen using transfection in non-lymphoid, mammalian cell lines^{18,50}. The rate was high enough to analyze events via flow cytometry and suggested that our yeast-based platform could be a viable alternative to mammalian cells for certain applications, such as studying defects in RAG1 and RAG2 proteins.”

And in the discussion, the following was added at line 839:

“Such demonstrations may be challenging given the relatively low rate of recombination. It is likely that a population could be enriched for coding joints prior to screening by using a negative selection, similar to the strategy we used to isolate signal joints. As recombination events

increase with time, allowing the RAG proteins more time to recombine their target should also improve the recombination rate.”

2. The abstract performs a metabolic engineering trick of reporting a fold amount (500-fold). This seems quite impressive until you note in the text that you are starting with essentially no recombination. I'd request that the authors report best recombination efficiencies in the abstract.

We agree with the reviewer that the “500-fold” number fails to communicate the actual rate. In responding to Reviewer #2's comments below, we generated additional results that show recombination rates approaching 1% after four days (**Figure 3f**). To achieve this, we combined the best homology-assisted recombination substrate (H20-U) and our best RAG1 truncation (348-1008) with RAG2 and HMGB1. To reflect this change, the fold increase was modified to be “over 7000-fold”. This is calculated from 1.33 CFUs/ 1×10^6 cells starting (R1c & R2c, **Figure 2b**) vs 9600 CFUs/ 1×10^6 cells final (**Figure 3f**). We now report the higher recombination efficiency alongside the ‘fold-increase’ number in the abstract.

The abstract was modified at line 10:

“By pursuing a variety of strategies, we increased the rate of homology-assisted recombination by over 7,000-fold, with the best rates approaching 1% recombination after four days.”

The final paragraph of the introduction was modified starting at line 90:

“We increased our initial recombination rate over 7,000-fold by applying codon optimization, varying protein combinations, adjusting RAG1 truncation, and optimizing the target substrate—ultimately reaching almost 1% recombination after 4 days.”

The following was added to the results section starting at line 590:

“We tested the best target plasmid with the highest-performing recombination strain we characterized earlier which uses RAG1(348-1006), RAG2, and HMGB1. RAG1core commonly ends at residue 1008, therefore we also built a similar strain which includes the additional two amino acids on the C-terminus (348-1008). Recombination rates reached nearly 10,000 CFUs/ 1×10^6 cells—or 1% recombination—after four days (**Figure 3f**). Unsurprisingly, we did not observe a significant difference between RAG1(348-1006) and RAG1(348-1008).”

And the first paragraph of the discussion was modified at line 779:

“By applying codon optimization, adding HMGB1, and using a truncated RAG1 and full-length RAG2, we achieved a homology-assisted coding joint recombination rate of nearly 1% after four days.”

3. I may have missed this in the discussion, but I didn't see the impact of temperature or of the functional reconstitution of these murine and human components on potential causes for low efficiencies. More generally, I also didn't see comparison with in vitro

reconstituted recombination. While outside of my field, I think the biochemistry and *in vitro* reconstitution of this system is mature and well worked out, and this literature may provide additional clues.

We are not aware of any published literature examining the effect of temperature on RAG expression or catalysis. This is an excellent point, and the effect on recombination of modifying temperature in yeast could be explored in future work. As the reviewer suggests, many studies have examined RAG1 and RAG2 behavior *in vitro*. This was usually accomplished by purifying the mouse core RAG proteins from mammalian, insect, or *E. coli* cells. To the authors knowledge, full-length RAG1 has to this day never been purified, highlighting the difficulty of working with the native sequence. It is difficult to directly compare *in vitro* and *in vivo* methods because *in vitro* techniques can only be used to examine DNA nicking or cleavage but not the joining events which are the usual *in vivo* readout. Nonetheless, our manuscript relies on many findings from *in vitro* studies such as encouraging us to begin with the mouse core proteins (as we note in the first paragraph of the Results section).

The following text modified/added at line 426:

“Low recombination levels seen with the full-length RAG pairings might be attributable to poor expression of the full-length proteins in yeast, particularly RAG1. This would agree with numerous *in vitro* studies of V(D)J recombination which have relied upon the core protein sequences for robust protein purification.”

Regarding the comparison between human and mouse sequences, we are not aware of studies which directly compare the activity/expression of these proteins, either *in vitro* or *in vivo*. In fact, the human proteins were only recently tested *in vitro* for the first time (reference Demirdjian *et al.* 2024), but it doesn't appear that the human proteins were exceptionally difficult to purify. Given that human core RAG1 and RAG2 can readily be purified, it suggests that their poor performance in yeast isn't due to an expression bottleneck. In addition, we now provide evidence that human RAG1core and RAG2 are expressed similarly to the mouse proteins in yeast (see our response to Reviewer #2's final comment). While we were unable to detect RAG1 expression in yeast using Western Blot, almost certainly due to an issue with the primary antibody, we could measure the expression level using eGFP tagging. Human RAG1core and RAG2 fluorescence was comparable to their mouse counterparts (**Supplemental Figure S7b**). A RAG2 Western Blot also confirmed that human RAG2 appeared to express similarly to mouse RAG2 (**Supplemental Figure S7c**).

The following text was added at line 502:

“While human RAG1core and RAG2core can be purified just like the more commonly used mouse proteins⁵⁴, we considered that the human proteins could suffer from poor expression in yeast. Like the RAG1 truncations, we checked the expression of human RAG1core and RAG2 by fusing each to GFP. Surprisingly, there was no significant difference in fluoresce between mouse and human RAG1core or RAG2 (**Supplementary Figure S7b**). To further validate the GFP tag data, we performed a western blot to directly measure the amount of RAG2 produced by the cells. Human and mouse RAG2 were both easily detected at comparable levels

(Supplementary Figure S7c). Therefore, our data indicate that the poor performance of human genes relative to mouse is not caused by deficient expression.”

A citation to the following publication was added to the manuscript:

<https://journals.asm.org/doi/10.1128/spectrum.02468-24>

4. The abstract statement below is overly broad, and ideally should be excised or otherwise constrained. What counts as a demonstration of *in vivo* generation of combinatorial genetic diversity? Wouldn't formation of functional antiviral RNA proteins using CRISPR arrays qualify? What about homologous recombination of different gene fragments in fungi and yeast?
“...which we believe to be the first ever demonstration of *in vivo* generation of combinatorial genetic diversity in a non-vertebrate cell.”

We thank the reviewer for providing feedback on this statement in the abstract, which we have adjusted. Our key point is that diversity is created entirely *in vivo* (i.e., at the time that diversity is created, no exogenous DNA/proteins/etc. are added/needed). As the reviewer notes, CRISPR arrays are diverse, but they require exogenous DNA/RNA (e.g., from a phage) to add to an existing array. Similarly, homologous recombination in yeast can certainly make hybrid/chimeric libraries, but when performed with intention as an engineering approach, it requires that DNA is carefully prepared and transformed into the cell. In terms of natural evolution via homologous recombination, we can't envision an example in which a single yeast population with a non-functional sequence naturally performs recombination to perform multiple different functional sequences. In our system, we take a monoclonal population of cells and introduce genetic and phenotypic diversity by simply inducing recombination events. We have modified our claim to better represent our work and the comparison we are trying to make.

The statement in the abstract has been modified at line 15:

“... first-ever generation of genetic and phenotypic diversity solely using random recombination of preexisting DNA in a non-vertebrate cell.”

In line with the reviewer's comment, our claims were similarly modified in the final paragraph of the introduction, line 99:

“In this manner, we showed in a non-vertebrate host cell the ability to study RAG1-associated immune deficiency and to generate *in vivo* combinatorial protein diversity via recombination of a predefined genetic locus.”

The first paragraph of the conclusion was modified as well, line 785:

“Finally, we used our platform to create the first instances of *in vivo* combinatorial diversity generation via recombination in yeast for two model protein classes: fluorescent proteins and antibodies.”

5. The inclusion of Figure 1 first is a bit confusing. If the point of figure 1 and 2 is the ultimate development of strain AC518, I would recommend combining the essential bits of figure 1 & 2 and porting the remainder to the SI. I also think adding sequences that are currently in the SI may improve the figures, but I'll defer to the authors here.

While Figure 1 does provide explanations for recombination rate data seen in Figure 2, we would argue it also has merits beyond the recombination assay. For example, the nucleolar sequestration of RAG1 had not been demonstrated in yeast previously. Moreover, the enhanced localization and comparable expression of RAG2 relative to RAG2core was not seen previously in yeast. While these may no longer represent the most essential findings of the manuscript overall, they helped explain engineering advances that were essential for the project's success, so we respectfully request that Figure 1 remain independent in the manuscript.

We are open to moving figures or sequences from the SI to a main figure, but we would ask for the reviewer's suggestions on which should be moved.

We again thank the reviewer for their time and effort helping to strengthen the quality of our manuscript.

Reviewer #2 (Remarks to the Author):

Cazier et al. report developing a yeast-based homology-driven V(D)J-like recombination system using engineered substrates and expressed RAG1 and RAG2 proteins. This idea of establishing ectopic RAG-mediated V(D)J recombination in yeast is not novel, which the authors acknowledge, but they do advance the reported efficiency of the recombination events by incorporating some clever ideas to optimize RAG expression and localization, incorporating homology-assisted repair, and also by including HMGB1/2 proteins. The system is highlighted for its potential utility in generating chimeric protein libraries. After establishing the baseline recombination activity with murine RAGs, they go on to evaluate how the system responds to human RAGs and mutant RAG proteins, as well as substrate mutations of various types (e.g. RSS mutations, intersignal distance, and combinatorial rearrangement). While there are some nice advances to the yeast RAG recombination system, there are several aspects of this work that make it more incremental than groundbreaking in its present form. These concerns are detailed below.

Major concerns:

1. A strength of the work is that the rearrangement events are pretty clearly RAG- and RSS-dependent. However, coding joint formation is not observed without homology-directed repair. The authors imply that because they observe signal joints, the reaction must involve classical introduction of double-strand breaks and NHEJ – the hallmarks of V(D)J recombination. However, there are no experiments to demonstrate that DNA breaks are actually generated via the known RAG-mediated nick-hairpin mechanism. Since the RAG proteins can nick DNA, the rearrangement events may instead be

progressing through sequential nicking without hairpin formation being an intermediate. Elegant assays have been done in the V(D)J recombination field to establish the cleavage mechanism, so similar experiments should be feasible. Furthermore, while the data establish consensus RSSs are required for recombination, the authors do not actually test whether the rearrangement follows the 12/23 rule in Fig. 3, which is another hallmark of V(D)J recombination. To show that, one would have to compare rearrangement frequencies between 12/23 and 12/12 or 23/23 recombination substrates. These are the major reasons that the work falls short of representing a significant advance to the field.

We thank the reviewer for these critiques/suggestions, and we feel that we have addressed them fully, and in doing so, greatly strengthened the key claims in the manuscript to help it more clearly represent a significant advance in the field. The reviewer highlights two important concerns regarding the mechanism of DNA cleavage in our yeast strains. **To address the first concern,** we now provide new results from additional RAG1 mutants described in previous work that have reduced functionality compared to the wildtype RAG1 protein. First, we tested R855A/R856A and R894A RAG1 mutants. RAG1 mutants R855A/R856A and R894A have been shown to be able to nick DNA adjacent to RSS sequences but lack the ability to form hairpins and thereby cleave DNA. Specifically, these mutants appear unable to correctly position the 3' OH group for transesterification and thus hairpin formation (reference Huye et al., 2002). Homology-assisted recombination activity was reduced over 500-fold for these variants. This shows that the mechanism of recombination in yeast likely proceeds through hairpin formation rather than sequential nicks, as postulated. Second, while not explicitly requested, we assayed D708N and E962Q, mutants for two of the three known catalytic residues which eliminate recombination activity but do not affect DNA binding. As expected, recombination was completely abolished. These results demonstrate that recombination is not caused by the RAG proteins binding or nicking DNA alone. For these experiments, all mutant strains used the same truncation of RAG1 (348-1008) and include full-length RAG2 and HMGB1. **These results are shown in Supplemental Figure S5a.**

The following paragraph was added to the results starting at line 471:

“To confirm that the homology-assisted recombination was reliant on the catalytic activity of RAG1, we constructed two strains harboring RAG1 mutants with essential catalytic residues mutated, either D708N and E962Q⁴⁹. While these catalytically dead mutants have been shown to have the ability to bind RSS sequences, recombination was completely abolished in these yeast strains, indicating RAG1 catalytic activity is required for homology-assisted V(D)J recombination (**Supplemental Figure S6a**). Next, we tested if RAG1 nicking (without double-strand cleavage) was sufficient to stimulate recombination. RAG1 mutants R855A/R856A and R894A have been shown to be able to nick DNA adjacent to RSS sequences but lack the ability to cleave DNA via hairpin formation⁵⁰. We created two strains that used these mutants, and RAG1 recombination was reduced over 500-fold for each strain (**Supplemental Figure S6a**). Therefore, yeast homology-assisted recombination appears to follow a similar DNA cleavage mechanism to that seen in mammalian cells.”

Citations to the following publications were added to the manuscript:

Cleavage-deficient mutants: <https://www.tandfonline.com/doi/full/10.1128/MCB.22.10.3460-3473.2002>

Catalytic mutants: <http://genesdev.cshlp.org/content/13/23/3059>

In generating this data, we note that we achieved our highest yet recombination rate of 1%. Given the significance of this result, it was added to the manuscript and is now referenced in the abstract and discussion. See also Reviewer #1, comment 2.

To address the reviewer's second concern, we created 12+12 and 23+23 target substrates (each with 20-bp homology between G418R fragments) to discern how well the homology-assisted recombination obeys the 12/23 rule. Compared to the original 12+23 plasmid, the 12+12 and 23+23 plasmids had almost no activity. This confirms that recombination is following the 12/23 rule, and suggests synaptic complex formation is required for DNA cleavage, as it is in mammalian cells. We retested the target plasmids with mutant RSSs alongside the 12+12 and 23+23 while generating this new data, so **the combined results are now shown in the updated Figure 3a**.

The following text was added at line 518:

“Furthermore, plasmids with two identical RSSs (12+12 and 23+23) also prevented recombination ...”

2. Another limitation is the lack of testing to see if hairpin nucleases could augment rearrangement efficiency, presuming that the coding ends are indeed sealed as DNA hairpin structures. While there are no direct homologs to Artemis and DNA-PK, it would be good to at least explore this question a bit more. Have the authors ever tried expressing constitutively active Artemis to see if that improves recombination activity? An activating truncation mutant has been described in the literature.

We thank the reviewer for this comment and suggestion. As the reviewer notes, when human Artemis is truncated (amino acids 1-413 of 692), it becomes constitutively active and does not require DNA-PKcs phosphorylation to open hairpin structures. We expressed Artemis(1-413) in a recombination strain (NLS-RAG1core, RAG2, and HMGB1) and performed a recombination assay with 20 bp of homology. While Artemis(1-413) did not significantly improve recombination efficiencies (**Supplemental Figure S4f**) in this instance, we agree with the reviewer that it may be important in future studies towards improving recombination activity, particularly when no homology-repair is possible. Therefore, future work will explore the potential for truncated Artemis to enable DNA repair without homology.

The following text was added to the results section, starting at line 491:

“One potential bottleneck for coding joint repair post-cleavage is the opening of coding end hairpins. In B cells, DNA-PKcs-phosphorylated-Artemis opens hairpins but a truncated Artemis (residues 1-413 of 692 total) has been shown to be constitutively active⁵³. We integrated yeast-codon-optimized Artemis(1-413) into strain AC518 and tested the rate of recombination.

Artemis(1-413) did not significantly increase the rate of recombination (**Supplemental Figure S6c**). It is possible that another yeast protein is already opening the hairpins effectively as part of the homology-directed repair.”

A citation to the following publication was added to the manuscript:

<http://www.jbc.org/content/281/45/33900>

3. Expression levels of the RAG proteins were never tested. Hence, it is unclear whether differences in expression can fully explain differences in recombination activity.

We agree with the reviewer’s comment, and we have taken every reasonable step to address this concern, which we feel we have now done. In general, our data show that higher RAG1 or RAG2 expression levels lead to higher recombination activity, but expression levels cannot fully explain the differences in recombination on their own. The original manuscript approximated expression level using GFP tagging at the C-terminus of many of the RAG proteins (**Figure 1a**). RAG1core expresses more highly than RAG1, and the recombination activity is clearly higher. In contrast, RAG2core appears to express more highly than RAG2, but the recombination activity is lower, likely due to reduced nuclear localization. Thus, expression cannot entirely explain the differences in recombination activity.

To reinforce these measurements and to examine the expression level in recombination-competent strains, we analyzed mouse RAG1 and RAG2 via western blot. Unfortunately, despite testing two unique commercial antibodies raised against different antigens in RAG1 and testing for presence of many RAG1 truncations/variants, we were unable to detect RAG1 expression in yeast with western blot. We were surprised by this result, as one prior group that expressed RAG proteins in yeast could measure RAG1 expression via western blot, though we note that they were able to do by incorporating a HA tag to the protein and then using a primary anti-HA mAb for their Western blots (reference Clatworthy *et al.*, 2003). Still, we were able to detect expression of RAG2 and show that RAG2core expresses similarly to RAG2 (**Supplemental Figure S7c**). The RAG2 results correlate with our previous GFP-tagging data. More description of our anti-RAG2 Western blots are included below in response to the reviewer’s next comment, and we will also mention that we used the same secondary antibodies for both Western blots, testing with either fluorophore or HRP conjugations formats. Please also see our response to the next comment for more details about RAG2 expression.

Because the primary western blot RAG1 antibodies we selected did not seem to work in our hands, we further examined the impact of RAG1 expression on recombination activity by tagging each of the RAG1 variants shown in Figure 2d with eGFP at the C-terminus (**Supplemental Figure S5**). We found that fluorescence correlates with recombination activity, but that the relationship is not strictly linear. Thus, we suggest that other factors such as localization, DNA-binding activity, or amenability to DNA repair are also influencing recombination activity.

The following paragraph was added to the manuscript starting at line 464:

“To elucidate the effect RAG1 truncation has on expression, we tagged each truncation with GFP and integrated them into yeast. Using flow cytometry, we measured the fluorescence of each construct (**Supplemental Figure S5a**). In general, higher expression correlates with higher recombination activity (**Supplemental Figure S5b**), but the correlation is not particularly strong, suggesting other factors such as localization or DNA-binding capability also impact recombination activity.”

Minor issues:

1. The authors report human RAGs perform poorly in the recombination assay, but it is unclear why that might be. Here again, expression data would be helpful in interpreting this result.

As the reviewer suggested, we also measured the protein expression levels of human RAG1 and RAG2 using western blot. Like mouse RAG1, we were unable to detect human RAG1 in yeast. However, human RAG2 appeared to express similarly to mouse RAG2 (**Supplemental Figure S7c**). We also measured the expression level using eGFP tagging. Human RAG1core and RAG2 fluorescence was comparable to their mouse counterparts (**Supplemental Figure S7b**). Therefore, we conclude that differences in expression are not likely to be the cause of poor human RAG recombination relative to mouse. Instead, innate differences in DNA binding or catalytic activity are more likely the cause.

The following text was added at line 504:

“Like the RAG1 truncations, we checked the expression of human RAG1core and RAG2 by fusing each to GFP. Surprisingly, there was no significant difference in fluorescence between mouse and human RAG1core or RAG2 (**Supplementary Figure S7b**). To further validate the GFP tag data, we performed a western blot to directly measure the amount of RAG2 produced by the cells. Human and mouse RAG2 were both easily detected at comparable levels (**Supplementary Figure S7c**). Therefore, our data indicate that the poor performance of human genes relative to mouse is not caused by deficient expression.”

We again thank the reviewer for their time and effort helping to strengthen the quality of our manuscript.

Generating combinatorial diversity via engineered V(D)J-like recombination in *Saccharomyces cerevisiae*

Andrew P. Cazier, Jaewoo Son, Sreenivas Yellayi, Lizmarie S. Chavez, Caden Young, Olivia M. Irvin, Hannah Abraham, Saachi Dalvi, and John Blazeck

Second Response to Reviewers

We thank the reviewers for examining our response, and we apologize for our erroneous statements in our response to Reviewer 1, and we are deeply thankful to Reviewer #2 for catching these oversights. As reviewer #1 was satisfied with our revisions, we only give our detailed response to the comments from reviewer #2.

RED – Reviewer comment BLACK – Author response

Reviewer #2 (Remarks to the Author):

The authors have performed several new experiments to address prior reviewer concerns. These include testing RAG1 "nick-only" mutants to indirectly show RAGs cut DNA in yeast through a "nick-hairpin" mechanism, testing recombination substrates containing 12/12 and 23/23 RSS pairs to show V(D)J recombination in yeast follows the "12/23 rule", and testing a constitutively active form of Artemis to determine whether recombination activity is stimulated by potential hairpin opening (no effect was seen). The authors also included experiments designed to evaluate RAG protein levels by immunoblotting. The addition of these experiments have strengthened the conclusions. Nevertheless, given the inefficiency of the recombination, and since there is no direct evidence that the "nick-only" mutants are actually only nicking DNA in yeast *in vivo*, the mechanism of cleavage remains indirectly tested. The authors should at least include a caveat stating this limitation in the discussion, and acknowledge that at present, hairpin opening, and hence the activity of Artemis, can also not be directly confirmed.

In the rebuttal letter, in response to Reviewer #1, the author makes some surprising statements regarding the state of the field regarding RAG biochemistry. The first is to suggest that full-length RAG1 has not been purified. However, Swanson's group in particular has purified full-length RAG1 and characterized its activity *in vitro* (e.g. PMIDs 14612436, 19233873, 22157821). Their finding in PMID 19233873 that full-length RAGs were slightly less active than core RAGs *in vitro* is in line with the authors findings. The second was the statement that "in vitro techniques can only be used to examine DNA nicking or cleavage but not the joining events which are the usual *in vivo* readout". However, the Gellert group has reported on cell-free V(D)J joining (PMID: 9242409). The authors should also be aware that the "nick-only" mutants are known to stimulate homologous recombination (PMID: 15084256). In that regard, it is noteworthy that these mutants have distinct activities in yeast, with R855A/R856A mutant showing no recombination, and the R894A mutant showing partial activity. The latter observation could in fact support the idea that RAG nicking, rather than RAG-mediated hairpin formation, is what is initiating recombination in this system. Since the relative rate of nicking between WT and R894A in this system is unknown, the fraction of recombination attributed to

RAG nicking could be quite high. This is an additional reason why a caveat statement is recommended.

We thank the reviewer for suggesting that a caveat be added to the discussion section regarding hairpin formation. We agree that our evidence for a hairpin intermediate in yeast RAG recombination is indirect.

The penultimate paragraph in the discussion section was substantially modified, starting at line 497:

“Although we were successful in generating combinatorial diversity, a weakness of our platform is that it does not create junctional diversity. The introduction of junctional diversity would most importantly require efficient production of coding joints via an error-prone repair pathway (i.e., not homology-assisted). Ideally diversity would be derived from both hairpin opening and non-templated polymerases, as occurs *in vivo*. We still lack direct evidence of hairpin formation and opening in yeast. While the results from the nick-only RAG1 mutants are consistent with the formation of a hairpin intermediate, adding constitutively active Artemis to a strain with wild-type RAG genes did not improve the rate of recombination as might be expected with a hairpin. If a hairpin intermediate is formed, it remains unknown what factors are responsible for opening it during HDR. Certain yeast proteins have been shown to possess nuclease activity with hairpin substrates, such as SAE2/MRE11 or PSO2⁶⁹⁻⁷¹. Hairpin opening in yeast may lead to end resection and commitment to an HDR pathway, which in turn would prevent error-prone, NHEJ-driven coding joint formation⁷². Similarly, it may be possible that NHEJ factors are being ‘outcompeted’ for coding-end binding by other, perhaps more prevalent, cellular proteins. In this vein, it may be important to bias DSB repair toward NHEJ through the deletion of various HDR genes⁷².”

We also moderated our claim in the body of the text at line 211:

Previous statement: “Therefore, yeast homology-assisted recombination appears to follow a similar DNA cleavage mechanism to that seen in mammalian cells.”

New statement: “Therefore, yeast homology-assisted recombination strongly benefits from a RAG1 that can efficiently cleave double-stranded DNA, similar to what is seen in mammalian cells.”

We thank the reviewer for highlighting two statements from the rebuttal which were incorrect. First, we apologize for missing the many demonstrations of full-length RAG1 purification, and we have added one of the suggested citations from the Swanson group in the body of the text.

The following sentence was modified, starting at line 159:

“This would agree with numerous *in vitro* studies of V(D)J recombination which have normally relied upon the core protein sequences due to the difficulty of purifying the full-length proteins³⁸.”

The following citation was added (PMID 14612436):

[https://www.jbc.org/article/S0021-9258\(20\)74980-X/abstract](https://www.jbc.org/article/S0021-9258(20)74980-X/abstract)

Second, the comment, “*in vitro* techniques can only be used to examine DNA nicking or cleavage but not the joining events...” was unintentionally worded too strongly. Nicking and cleavage events can be directly detected *in vivo* with electrophoresis/blotting techniques and joining events can be accomplished *in vitro* when cell extracts are supplemented to the purified RAG proteins. A citation to the Gellert group’s work showing cell-free recombination was already included in the manuscript at line 181.

Regarding the mechanism of repair, other avenues may be explored in the future to see if hairpins are formed. In particular, we are pleased to share that (very) recent (and very preliminary) work in our lab has shown that coding joints can be created in yeast without any homology between the coding ends. The current rate is very low, making it difficult to even quantify, but sequencing has shown that the joints are somewhat diversified and often include small insertions (e.g., a single added codon). It is possible that a comprehensive analysis of such joints will reveal the presence of p-nucleotides, a hallmark of hairpin intermediates in V(D)J recombination. We anticipate that such an analysis will be included in a future publication.

We again deeply thank the reviewer for their thoughtful consideration of our work.